# Identifying urban and rural settlement archetypes: clustering for enhanced risk-oriented exposure and vulnerability analysis

Gabriella Tocchi[a], Massimiliano Pittore[b], Maria Polese[a]

[a] *University of Naples Federico II, Department of Structures for Engineering and Architecture, Naples, Italy*
[b] *EURAC Research, Center for Climate Change and Transformation, Bolzano, Italy*

*Correspondence to*: Gabriella Tocchi (gabriella.tocchi@unina.it)

**Abstract.** Identification of risks and vulnerabilities in urban and rural areas is crucial for supporting local authorities in disaster risk reduction and climate change adaptation. Moreover, comparison of risk assessments across different areas may help effective allocation of adaptation funding towards more resilient and sustainable communities. The distinct physical, social, economic, and environmental characteristics of a settlement, along with the relevance of impending hazards, determine the level of risk and vulnerability faced by its residents. While the results of risk assessments will vary from one settlement to another, using general settlement typologies (e.g. coastal cities, dryland cities, and inland or high-altitude cities) can effectively support the understanding of risk in relation to its key drivers, helping to segmentate the complexity in an otherwise too broad problem (Dickson et al., 2012).

This study aims to reduce complexity in risk assessment of urban/rural settlements at regional and national scale, ensure a baseline for comparison and identify potential hotspots in risk assessment frameworks. We propose a clustering methodology that groups human settlements based on open-source data, used as proxies of urban vulnerability and exposure. Applying two widely used clustering techniques, we define 18 urban and rural archetypes for the Italian territory, incorporating geographic, demographic, and socio-economic characteristics. These archetypes satisfy multiple validity dimensions of archetype analysis (Piemontese et al., 2022) and can serve as a valuable tool for policymakers. By providing a structured understanding of human settlements vulnerability profiles, they support the design of targeted interventions and resilience strategies tailored to specific risk conditions.

## 1    Introduction

Over the last few decades, natural disasters have caused devastation to many communities throughout the world, killing about 1.5 million of people and incurring losses exceeding 4.5 billion USD (Centre for Research on the Epidemiology of Disasters - CRED, 2024). Such disasters are the results of the interaction of hazards (natural or man-made) with vulnerable socio-ecological and socio-economical systems. Evidence shows that the level of disaster proneness of communities may vary greatly with their physical, demographic, socioeconomic and institutional characteristics (Cutter et al., 2003; Wang et al., 2022). For example, low-income and minority communities in New Orleans were disproportionately affected during Hurricane Katrina due to residing in flood-prone, lower-lying areas, and lacking personal transportation, which hindered evacuation (Flanagan et al., 2011). Similarly, aging communities with limited mobility face challenges in evacuating quickly during hazardous events, leading to higher mortality rates, as seen during the 2011 Tohoku Tsunami, Hurricane Katrina, and the 2017 and 2018 California wildfires (Brunkard et al., 2008; Hamideh et al., 2022; Miyazaki, 2022).

Climate change brings additional challenges to management and decision making for city governments and is associated with a growing variety of impacts on cities, the surrounding ecosystems, and livelihood of resident and temporary population (e.g., Dickson et al., 2012). As highlighted in the IPCC's 6th assessment report, in urban areas the risk to people and assets due to climate-related hazard has already increased, and climate impacts are felt disproportionately in urban communities, with the most economically and socially marginalized being most affected (Dodman et al., 2023). Such risks depend on the increase of intensity and frequency of extreme weather events (La Sorte et al., 2021; Mulholland & Feyen, 2021) as well as on the interplay with several non-climatic risk drivers including extent and features of the exposed systems and assets (e.g., European Environment Agency, 2024) and their vulnerability (e.g., Cutter & Finch, 2008; Dickson et al., 2012).

Exposure refers to the presence of people, livelihoods, species or ecosystems, environmental functions, services, and resources, infrastructure, or economic, social, or cultural assets in places and settings that could be adversely affected, while vulnerability refers to the propensity or predisposition to be adversely affected. Vulnerability encompasses a variety of concepts and elements, including sensitivity or susceptibility to harm and lack of capacity to cope and adapt (Intergovernmental Panel on Climate Change - IPCC, 2022; Koren et al., 2017). It encompasses both the lack of coping

capacity and adaptive capacity—factors that influence a community's ability to manage disasters effectively (Cardona et al., 2012; Marin Ferrer, 2017). The level to which urban settings are prone to the negative impacts of one or multiple hazards is also known as urban vulnerability (Thywissen, 2006), and its assessment is particularly challenging, as cities are intricate systems composed of interdependent networks of built environments, infrastructure, and social systems (Koren et al., 2017). The concentration of assets and people may increase potential losses, while dynamic interactions between individual components that enable efficient system performance can lead to cascading failures. In addition, urban areas are often exposed to multiple hazards, such as earthquakes, floods, heatwaves, each interacting with the built environment and human activities in different ways. Rural settlements, on the other hand, may experience different forms of vulnerability, often related to geographic isolation, limited access to emergency services and infrastructure, lower institutional capacity, and demographic challenges such as aging population, which can significantly hinder preparedness and recovery. This complex interplay explains also why often non-extreme hazards can lead to severe consequences, while extreme events in other contexts may not result in disasters (Lavell et al., 2012).

In this complex context, archetypes can be powerful tools for simplifying and interpreting systemic risks They provide structured representations of recurrent patterns across diverse cases, helping policymakers understand key drivers of vulnerability and exposure and supporting more effective risk communication and decision-making (Oberlack et al., 2023; Piemontese et al., 2022; Wicki et al., 2024). Archetypes have been extensively employed to classify cities based on socio-economic and socio-demographic parameters, to support policy decisions on fiscal interventions (Bruce, 1971; Dalton, 2015; Harris, 1943). An increasing amount of climate studies are dedicated to identifying recurring patterns and archetypes, in order to understand local climate vulnerabilities and to formulate specific adaptation strategies (Rocha et al., 2020; Vidal Merino et al., 2019; Wicki et al., 2024). For instance, in Riach et al. (2023) recurring climate risk patterns at the municipal level in Baden-Wuerttemberg, Germany, are identified by analysing indicators for climatic hazards (e.g., annual mean temperature, hot/ice days, heavy precipitation) and exposure/vulnerability (e.g., proportion of elderly, energy production, population density). The nine urban archetypes derived represent municipalities with varying climate risk characteristics that require tailored adaptation measures. Although several examples of city-scale archetypes analysis are available, they often focus on the analysis of single-hazard risk (e.g., Awah et al., 2024; Carroll & Paveglio, 2016; Joshi et al., 2022; Riach et al., 2023) and may be not applicable in a multi-risk context.

This study addresses the following research question: *can urban and rural settlements be clustered into meaningful archetypes based on shared characteristics of vulnerability and exposure, to improve multi-risk assessment and support more targeted resilience planning at regional and national scale?* Indeed, despite the high specificity of exposure and vulnerability of each urban and rural environment, we assume that a relatively low number of representative archetypes could be found to decrease the level of complexity at regional and national scale, ensure a baseline for comparison and highlight potential hotspots in multi-hazard and multi-risk assessment frameworks.

The term "archetype" can be interpreted in different ways. In statistics, archetypes refer to extremal profiles used to describe all data points as convex combinations of a few "pure" types (Cutler & Breiman, 1994). In contrast, in sustainability science and climate risk research, archetypes are understood as representative specimens or clusters of similar entities that are "crucial for describing the system dynamics or causal effect of interest" and that exhibit recurring patterns of risk-relevant characteristics " (Oberlack et al., 2019). We adopt this latter interpretation. In our work, urban and rural settlement archetypes are defined as representative instances (real or ideal) of a group of municipalities sharing similar vulnerability and exposure characteristics.

Following the approach suggested in Piemontese et al. (2022), we perform the archetype analysis in Italy according to three phases of design, analysis and application. In the *design phase*, the problem framing and attributes selection is performed. In particular, this study seeks to address the challenge of assessing urban/rural exposure and vulnerability by proposing a national-scale clustering of Italian settlements using open-source data. Municipality is selected as the primary geographical boundary for settlements since available authoritative open-source data is often referring to such administrative units. Municipalities are small, well-defined units, making them ideal for detailed spatial analysis and accurate identification of human settlements. These boundaries often reflect historical settlements, preserving the cultural context that is essential for understanding contemporary urban dynamics. Additionally, municipalities are responsible for local governance and urban planning, making them relevant units for studying urban/rural settlements, as local policies directly affect development and quality of life (actionability also for risk mitigation and climate change adaptation). The goal of this study is to group settings (municipalities) to define risk-oriented urban and rural settlements archetype. To this end, we select a set of geographic, demographic, and socio-economic attributes available from open-source data, known to be relevant to vulnerability/resilience (see section 2). Thanks to a proper selection of a range of geographic, demographic, and socio-economic parameters, the study provides a robust assessment of the vulnerability of Italian urban and rural settlements, identifying archetypes with varying levels of susceptibility to natural hazards. Moreover, the use of

open-source data ensures the approach is both replicable and scalable, making it generalizable and applicable to other regions. For the *analysis phase*, described in section 3, methods of analysis should be defined towards generalizability of results. Archetypes are derived through a two-step clustering process: first, broad urban and rural archetypes are defined using only demographic and geographic data, then they are refined using socio-economic attributes. This initial classification reduces complexity and establishes a baseline for comparison, providing a clear, interpretable framework to capture essential structural differences among urban/rural settlements (e.g., size, density, location). Refining these archetypes with socio-economic parameters allows for a more articulated understanding of vulnerability differences within similar structural contexts, supporting more targeted risk assessment and policy intervention. This two-step approach balances clarity with detail, enhancing both usability and precision. The proposed methodology utilizes two widely-used clustering techniques—*agglomerative hierarchical clustering* and *partitioning clustering*—to analyse vulnerability-related data. Using two clustering techniques allows for cross-validation of results and helps capture different patterns in the data, enhancing the robustness and reliability of the identified archetypes. Results of the cluster analysis are presented in sections 4 and 5. Finally, the *application phase* entails the practical usefulness and a real-world check of the archetypes, meaning they should correspond to variable levels of susceptibility to risk (according to the problem framing), and assessment of the impact, intended as the usefulness of results for application by final knowledge users. To this aim, a simplified Impact Susceptibility Index is proposed, highlighting the likelihood of experiencing negative consequences based on the combined levels of vulnerability and exposure associated with each identified archetype. Additionally, Section 7 provides a comprehensive discussion on how each dimension of archetype analysis validity - as outlined by Piemontese et al. (2022) - is addressed, emphasizing both the strengths and limitations of the study.

By developing a national-scale clustering of Italian municipalities, 10 broad and 18 nested archetypes are identified in this study. The identified archetypes offer a simplified framework for managing the complexity of diverse areas and their exposure to hazards. This risk-oriented classification offers valuable insights for resilience and disaster management professionals, enabling policymakers and urban planners to design targeted risk-reduction strategies tailored to the specific vulnerability profiles of each archetype, resulting in more efficient resource allocation.

## 2    Selection of key indicators of vulnerability dimensions

To apply clustering techniques, it is essential to have a dataset containing meaningful features (attributes) that allow for clear differentiation between clusters. These attributes may include numerical, categorical, or mixed data types, depending on the clustering algorithm. Thus, the first step in clustering human settlements at a national scale is to identify key drivers of vulnerability and assess data availability.

Vulnerability is multidimensional, defined by various physical, social, economic, environmental, and institutional factors that shape the susceptibility of systems to the impact of hazards (UNDRR., 2023; Van Westen & Woldai, 2012; Villagrán de León, 2006). Social vulnerability refers to the propensity of some social groups (e.g., poor, single-parent households, pregnant or lactating women, the handicapped, children, and elderly) to suffer negative consequences of hazards, due to their lack of capacity to react and manage the effect of hazard related processes (Cutter et al., 2003; Wisner et al., 2004). Economic vulnerability is the propensity of economic assets and processes to be harmed by exogenous shocks (Cardona et al., 2012), such as the potential impacts of natural and man-made hazards (i.e., business interruption, secondary effects such as increased poverty and job loss). Physical vulnerability expresses the propensity of the built environment (e.g., buildings and infrastructure) and population to suffer the physical impact of hazardous events (Douglas, 2007). Institutional vulnerability arises from limitations in governance structures, risk communication, preparedness, and emergency management systems. Following Papathoma-Köhle et al. (2021), institutional vulnerability also encompasses the capacity of institutions to anticipate, absorb, and adapt to hazards, highlighting the importance of coordination, contingency planning, and learning mechanisms as part of adaptive risk governance. Environmental vulnerability is the susceptibility of ecosystems to sustain degradation (destruction of forest, farmland, or crops, lower yields) and loss of functionality following a hazardous event (Angeon & Bates, 2014; Marzi et al., 2019). Table 1 presents a list of key indicators commonly used in literature to assess each dimension of vulnerability mentioned.

In our work we focused on a selection of indicators, expectedly linked with different vulnerability dimensions, and namely: altimetric zone, centeredness degree, degree of urbanisation, residential population and social vulnerability indicators. The altimetric zone of a settlement, which refers to their elevation and topographical features, can be considered a proxy of access to the main services – or equally distance to services centres (institutional vulnerability, see Table 1). Accessibility of services of general interest can be particularly challenging in certain contexts (e.g. mountain regions, islands) due to their geomorphological and settlement structure conditions (Bertram et al., 2023). These

accessibility issues can also complicate evacuation efforts and the delivery of emergency services during a disaster. Likewise, degree of urban centeredness, which reflects the spatial characteristics and distribution of urban areas, is associated with the availability of public services and the level of spatial connectedness, as it measures the distance and travel time to major service centres (institutional vulnerability, see Table 1). The degree of urban centeredness significantly influences the response and resilience of urban systems by affecting resource availability, infrastructure robustness, community networks, and emergency preparedness (Giuliano & Narayan, 2003; Schwanen et al., 2004). Ensuring effective access to essential public services, such as healthcare and education, is challenging even under normal circumstances. However, it becomes even more crucial during crises like natural disasters, when the demands on these services and their operating conditions become significantly more complex (Fan et al., 2022; Loreti et al., 2022; Tariverdi et al., 2023). The level of peripherality of the areas with respect to the network of urban centres influence may determine not only difficulties of access to basic services but also lower quality of life of citizens and their level of social inclusion (Oppido et al., 2023).

**Table 1 – Vulnerability dimensions most common indicators.**

| Dimension | Indicator | Reference |
|---|---|---|
| Social | Dependency ratio | (Cutter et al., 2003; Eriksen & Kelly, 2007; Frigerio et al., 2018) |
| | Age | (Cutter et al., 2003; Frigerio et al., 2018; Marzi et al., 2019) |
| | Population growth | (Cutter et al., 2003; Frigerio et al., 2018) Fare clic o toccare qui per immettere il testo. |
| | Level of education | (Cutter et al., 2003; Frigerio et al., 2018; Marzi et al., 2019; Sibilia et al., 2024) |
| | Family structure | (Cutter et al., 2003; Frigerio et al., 2018; Marzi et al., 2019) |
| | Commuting rate | (Cutter et al., 2003; Frigerio et al., 2018; Marzi et al., 2019) |
| | Quality of buildings | (Cutter et al., 2003; Frigerio et al., 2018) |
| | Race/Ethnicity | (Cutter et al., 2003; Frigerio et al., 2018; Marzi et al., 2019) |
| | Access to medical services | (Cutter et al., 2003; Sibilia et al., 2024) |
| Economic | Employment rate | (Marzi et al., 2019; Opach et al., 2020; Sibilia et al., 2024) |
| | % women in the workforce | (Marzi et al., 2019; Opach et al., 2020) |
| | Household income | (Marzi et al., 2019; Sibilia et al., 2024) |
| | GDP per capita | (Eriksen & Kelly, 2007; Sibilia et al., 2024) |
| Physical | Housing conditions | (Marzi et al., 2019; Sibilia et al., 2024) |
| | Building typology/material/design | (FEMA, 2022; Kappes et al., 2012; Lagomarsino & Giovinazzi, 2006) |
| | Population density | (Marzi et al., 2019; Opach et al., 2020) |
| Institutional | Access to services/ Distance to services centres | (Marzi et al., 2019; Opach et al., 2020) |
| | Political stability | (Papathoma-Köhle et al., 2021; Sibilia et al., 2024) |
| | Risk awareness and perception | (Papathoma-Köhle et al., 2021) |
| | Transparency | (Papathoma-Köhle et al., 2021) |
| Environmental | Vegetation cover/Land use | (Eriksen & Kelly, 2007; O'Brien et al., 2004; Sibilia et al., 2024) |
| | Water quality and availability | (Eriksen & Kelly, 2007; Rockstrom, 2013) |
| | Air pollution level | (Cohen et al., 2017; Eriksen & Kelly, 2007) |

Residential population and degree of urbanitation are linked to exposure and physical vulnerability dimensions, and specifically to population density (physical vulnerability, see Table 1). While population density cannot capture the full range of structural vulnerability factors of the built environment, it reflects both the intensity of exposure and the systemic vulnerabilities inherent to high-density urban environments (e.g., emergency response complexity and evacuation challenges, increased likelihood of cascading infrastructure failures during hazard events, overburdened urban services that exacerbate systems' physical fragility - healthcare, water systems, mobility - under stress), consistent with its interpretation in urban risk literature (e.g., Balk et al., 2018; Marzi et al., 2019; Opach et al., 2020; Zhao et al., 2017). Residential population significantly influences the exposure to natural hazards, determined not only by the higher presence of people and housing, but also of infrastructure, production capacities, species or ecosystems, and other tangible human assets in places and settings that could be adversely affected by one or multiple hazards. Greater population not only increases the potential for human and property losses, but also complicate evacuation efforts, and strains emergency response resources (Zhao et al., 2017).

The degree of urbanization is often used to classify areas into cities, urban areas, and rural areas based on criteria such as population density, concentration of human activities, and built environment (Balk et al., 2018; United Nations, 2018). Indeed, highly urbanized densely populated areas are more likely to experience greater damage, congestion, and strain on resources during emergencies. It affects the capacity for evacuation and accessibility to essential services, due to dense infrastructure, complex urban layouts and the potential for cascading failures in infrastructure (Kendra et al., 2008; Lall & Deichmann, 2012).

Finally, social vulnerability indicators include those parameters that influence both social and economic vulnerability. Past events highlight that the elderly may be more vulnerable due to reduced mobility, poor health, and communication challenges (Ardalan et al., 2010; Carnelli & Frigerio, 2017; Cutter et al., 2003), while education levels can heighten vulnerability to natural hazards influencing risk perception and awareness, knowledge, and skills related to disaster preparedness (Alexander, 2012; Wachinger et al., 2013). Still, minority groups, including migrants, and ethnic communities, often face heightened social vulnerability, especially in high-risk areas, due to language barriers and communication challenges that can hinder access to critical emergency information (Carnelli & Frigerio, 2017; Walter Gillis et al., 2012). A comprehensive list of socio-economic indicators considered is presented in section 2.5, though some indicators were not used for the final clustering process due to their strong correlation with other selected variables (Table 2).

It is worth mentioning that we only consider indicators for which publicly available and authoritative data exist at the municipal level. For example, since GDP per capita is only available at national, regional, or provincial scales, it is not included in this study. Similarly, many building characteristics affecting physical vulnerability are either difficult to detect or unavailable at the municipal scale (e.g., structural system and earthquake-resistant design level; Tocchi et al., 2022). Moreover, building vulnerability indicators often vary depending on the type of hazard (Kappes et al., 2012), making it challenging to collect all relevant information for multiple hazards across Italy. For these reasons, only population density and general building quality are considered in this study. Indicators suggested for the environmental vulnerability dimension are not included due to data limitations as well. For instance, municipal-level air pollution data in Italy is limited, as such data is only available for major cities with monitoring stations.

Data on degree of urbanisation, the degree of urban centeredness, altimetric zone, social vulnerability factors used herein are primarily sourced from ISTAT (Italian National Institute of Statistics). All data are collected at the municipal level, aligning with the administrative boundaries adopted for the analysis. The dataset includes 7960 objects, representing the 7960 Italian municipalities, and 19 attributes (both numerical and categorical) related to the vulnerability factors outlined in sections 3.1 through 3.5.

## 2.1    Degree of urbanization

Eurostat (2021) proposed a grid-based approach, implemented in geographic information systems (GIS), to determine the degree of urbanization based on a combination of geographical contiguity and population density. First, raster grid cells of 1 km² are categorized according to their total population and population density. Second, small statistical units are classified as urban centres (high-density units), urban clusters (moderate-density units) and rural cells (low density units) based on population thresholds and density criteria of groups of neighbouring cells. Finally, degree of urbanization of local administrative units is defined based on the share of population living in urban centres, urban clusters and rural grid cells: densely populated units (i.e., at least 50% of population living in urban centres) are categorized as *cities*, thinly populated units (i.e., at least 50% of population living in rural areas) as *rural areas* and intermediate density units (i.e., less than 50% of population living in rural areas and urban centres) as *towns and suburbs*. In Italy, classification of municipalities adopting the above-mentioned Eurostat procedure is provided by ISTAT, the Italian National institute of statistics (https://www.istat.it/classificazione/principali-statistiche-geografiche-sui-comuni/) and reported in Figure 1.

It is found that only 3% of Italian municipalities are classified as cities, yet they account for 33% of the country's residential population. Conversely, rural areas make up 68% of municipalities but only represent 24% of the population. Towns and suburbs, which comprise 29% of municipalities, account for 43% of the population.

## 2.2    Degree of urban centeredness

The degree of urban centeredness is used herein as a measure of how centralized or decentralized an urban area is. Italian territory is a polycentric territory, i.e., a territory characterized by a network of municipalities or aggregations of municipalities (centres of offer of services) around which areas characterized by different levels of peripherality gravitate. These centres offer a wide range of essential services, capable of generating important catchment areas, even remotely, and of acting as "attractors" (in the gravitational sense). The methodology used to define the degree of urban

centerednessof municipalities is based on the approach proposed by the National Strategy for Internal Areas (SNAI, "Strategia Nazionale per le Aree Interne" in Italian). This territorial policy aims to enhance the quality of citizen services and economic opportunities in remote areas, which are characterized by significant distances from major service centres and are at risk of marginalization. The proposed methodology involves two main phases: i) identifying urban hubs based on their capacity to provide essential services and ii) classifying the remaining municipalities as peri-urban areas and

inner areas based their distance from the hubs measured in travel time (DPS, 2013).

More specifically, the selection of hubs, which can also be defined as service offering centres, is based on service availability indicators for high school educational services (e.g., high schools, technical and professional institutes, and other higher education institutions), health services (e.g., presence of multiple health and emergency facilities, healthcare facilities with at least 250 beds), and rail transport services (e.g., train stations with an average of more than 6,000

travellers per day and a high number of daily trains). Some neighbouring municipalities are classified as intermunicipal hubs, meaning that several contiguous municipalities collectively provide the required level of services in a network system. The remaining municipalities are classified based on an accessibility indicator measured in minutes to reach the nearest hub. Peri-urban areas are less than 20 minutes away from the nearest hub, while inner areas are more than 20 min away. Further classification of the inner areas into three categories is also provided: it is possible to distinguish between

intermediate areas, that are approximately 20 to 40 minutes away, peripheral areas, that are between 40 and 75 minutes away, and ultra-peripheral territories, that are more than 75 minutes away. Figure 1 (right) shows the classification of Italian municipalities based on their degree of urban centeredness.

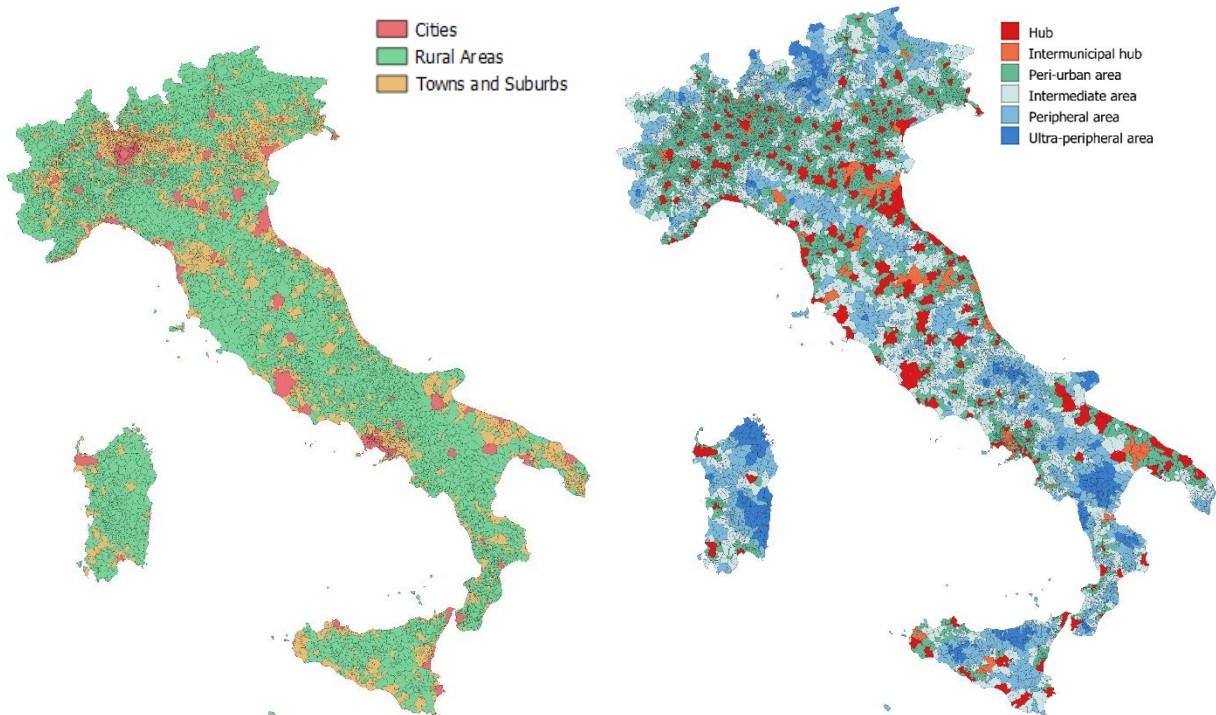

**Figure 1 – Italian municipalities classified based on degree of urbanisation (left) and degree of urban centeredness (right).**
**Data used for the classification are derived from ISTAT (https://www.istat.it/classificazione/principali-statistiche-geografiche-sui-comuni/).**

Most of the Italian population resides in peri-urban areas (37%) and urban hubs (35%), which account for 44% and 3% of municipalities, respectively. Intermediate, peripheral and ultra-peripheral areas account for 16%, 6% and 1% of population, respectively, and represent 28% (intermediate), 19% (peripheral), and 4% (ultra-peripheral) of Italian

municipalities. Intermunicipal hubs represent only 2% of municipalities and house 5% of the population. Population density generally decreases from hubs to peripheral municipalities. High-density cities comprise 35% of hubs, while medium-density towns and suburbs make up 57%. Only 8% of hubs are low-density rural areas. Intermunicipal hubs exhibit medium-high population density, with 23% classified as cities, 50% as towns and suburbs and 27% as rural areas. Peri-urban municipalities exhibit medium-low population density, with 5%, 45% and 50% classified as cities, towns and

suburbs and rural areas, respectively. Intermediate, peripheral, and ultra-peripheral municipalities are mostly low-density rural regions (83%, 91%, and 96%, respectively).

It is worth mentioning that only three classes are considered for the degree of urban centeredness, namely urban hubs (represented by both hubs and intermunicipal hubs, shades of red in Figure 1), peri-urban areas (green in Figure 1) and inland areas (that includes intermediate, peripherical and ultra-peripherical areas, shades of blue in Figure 1), according to the main classification proposed by ISTAT. This simplification is adopted in order to: (i) minimize noise and variability in the data, leading to more stable and reproducible clusters; (ii) prevent the model from overfitting to minor variations, improving generalizability; (iii) enhance interpretability.

## 2.3  Altimetric zone

ISTAT classifies Italian municipalities into three altimetric zones based on elevation: mountain (>600 m.a.s.l.), hill (300-600 m.a.s.l.), and lowland (<300 m.a.s.l.) (ISTAT, 2020). Elevation data is derived from a Digital Elevation Model (DEM) developed by ISPRA (Italian Institute for Environmental Protection and Research) for a 20-meter grid. Using the DEM, statistics such as average, sum, minimum, and maximum elevation within the municipal boundaries are calculated using a zonal statistics tool in GIS software. The municipality's altimetric zone is then determined based on the surface prevalence criterion. Municipalities could be also subdivided to account for the moderating influence of the sea on the climate, as coastal or inland areas. However, only the main three altimetric classes are adopted in this study (i.e., mountain, hill and lowland), not considering the classification in coastal and inland zones. Indeed, coastal zones represent a minority of municipality (almost 90% of municipalities are located in inland areas, while only 10% in coastal areas) and this can lead to the model becoming more likely to fit to noise, reducing its generalizability to new data. Furthermore, despite some correlation existing between urban vulnerability and coastal/inland areas (e.g., a coastal city with strong infrastructure and preparedness may be less vulnerable than an inland town with weak governance and high poverty), generally the distinction between coastal and inland areas is primarily linked to the types of natural hazards affecting these regions rather than inherent differences in urban vulnerability. Figure 2 shows the classification of Italian municipalities by population classes and altimetric zones.

Geographically, 31% of municipalities are in mountainous areas (30% inland, 1% coastal), accounting for just 12% of the population (10% inland, 2% coastal). Hill areas encompass 43% of municipalities, with 33% in inland hills and 10% in coastal hills, representing 23% and 16% of the Italian population, respectively. Lowland areas include 26% of municipalities (24% inland, 2% coastal) and home to 49% of the population (34% inland, 15% coastal).

Many densely populated cities are located in lowland areas (75%), while only 22% are situated in hilly regions and 3% in mountainous areas. Low-density or rural municipalities are predominantly found in mountainous (39%) and hilly regions (42%), compared to only 19% in lowlands. Medium-density towns and suburbs are mostly located in lowlands (38%) and hills (45%), with only 17% in mountainous areas. Similarly, hubs (including intermunicipal hubs) and peri-urban municipalities are primarily situated in lowland (47% and 44%, respectively) and hilly areas (44% and 41%), with only 9% of hubs and 15% of peri-urban municipalities in mountainous regions. In contrast, most inner municipalities are in mountainous (47%) and hilly regions (44%), while only 9% are found in lowlands.

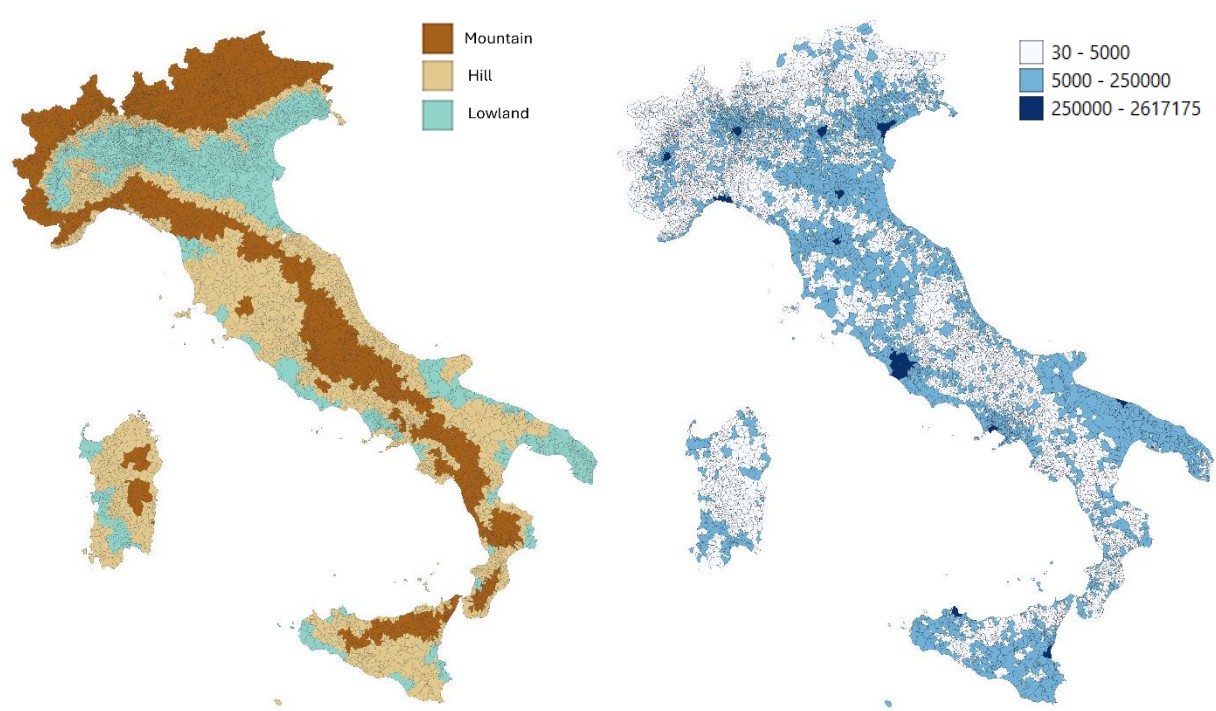

**Figure 2 - Italian municipalities classified based on altimetric zone (left) and population (right). Data used for the classification are derived from ISTAT (https://www.istat.it/classificazione/principali-statistiche-geografiche-sui-comuni/).**

## 2.4 Residential Population

Three population classes ($C_{pop}$) were introduced by ISTAT to classify municipalities according to the number of
300 inhabitants (ISTAT, 2020). The classes (Figure 2) are defined using the following population thresholds: small municipalities (less than 5000 inhabitants, $C_{pop}=1$); medium municipalities (between 5001 to 250000 inhabitants, $C_{pop}=2$); big municipalities (more than 250000 inhabitants, $C_{pop}=3$). ISTAT provides updated statistics on the resident population per municipality every year. In this study information on population per municipality is updated to 2018, along with the most recent data on urbanization, centrality, and altitude zones, all referring to the same year.

A significant proportion of municipalities fall into the lowest population class, with 33% having between 501 and 2000 inhabitants, 26% having between 2001 and 5000 inhabitants and 11% being very sparsely populated, with fewer than 500 residents. The remaining municipalities belong to the medium population class (30%), including 15% with population between 5001 and 10000 inhabitants, 13% between 10001 and 50000, and 2% between 50001 and 250000. Only 0.2% belong to the high population class, representing Italy's largest cities such as Rome and Milan.

As expected, 91% of hubs (including intermunicipal hubs) are found in higher population classes, with 87% in population class 2 and 4% in population class 3, highlighting their concentration in highly populated areas. In contrast, 84% of inner municipalities fall within the lowest population categories, with only 16% classified in class 2. Similarly, densely populated cities tend to have larger populations, with 73% in classes 2 and 3 (41% having between 10001 and 50000 inhabitants, and 32% exceeding 50001 inhabitants). Meanwhile, 88% of rural areas are also sparsely populated, with the
vast majority (88%) having fewer than 5000 inhabitants.

## 2.5 Social vulnerability indicators

Parameters commonly used to assess social vulnerability, such as gender, age, education, socioeconomic status, public health condition, employment status, and access to resources, need to be tailored to the local context to accurately reflect place-specific dimensions (Chen et al., 2013; Cutter et al., 2003; Guillard-Gonçalves et al., 2015; Mesta et al., 2022). The
320 variables representing social vulnerability adopted in this study are derived from the study conducted by Frigerio et al (2018). These variables encompass seven demographic and socio-economic indicators pertinent to the Italian context. and specifically: age, population growth, level of education, family structures, commuting rate, quality of buildings, race/ethnicity and employment. Age indicator includes the percentage of children (under 15) and elderly (over 65), the ageing index, calculated as ratio of elderly to children (Preedy & Watson, 2010), and the dependency ratio, i.e., ratio of

nonworking-age people to working-age people (Simon et al., 2012), calculated in this study as those under 15 and over 65. The family structure indicator measures the proportion of families with more than five members. Indeed, evidence shows that the larger the family the lower the income (ISTAT, 2024). The education indicator consists of the low educational index, calculated as number of people with at most a secondary school diploma compared to the total population aged over 15, and the high educational index, calculated as people with at least a university degree compared to the total population aged over 30. The commuting rate is the ratio of commuters to working-age people (those over 15), while building quality indicator is calculated as the proportion of buildings in poor condition. In the census database, building quality is classified based on four categories of preservation: very good, good, bad, or very bad. For this study, the number of buildings in bad or very bad condition at the municipal level is used as a representative variable for building quality. The employment indicators cover both unemployment, employment, and female employment rates among working-age people (those over 15). The crowding index is calculated herein as the number of persons per dwelling. The race/ethnicity indicator is defined in terms of percentage of foreign population (i.e., not Italian citizens). It is important to note that only 12 of the 14 previously presented social vulnerability indicators are used in this study (Table 2), as a correlation analysis - described in Section 3.1 - was conducted. Each variable is derived from last census (ISTAT, 2011) at census tract level and aggregated at municipal level.

It was observed that aging index, dependency ratio, low educational index and the percentage of buildings in poor conditions tends to increase from hubs to ultra-peripheral areas (average values of 1.7, 0.55, 0.51 and 14% for hubs, average values of 2.7, 0.61,0.60 and 20% for ultra-peripheral areas, respectively), while high educational index, percentage of employed and female employed as well as crowding index tend to decrease (average values of 0.17, 50%, 44% and 2.4 for hubs, average values of 0.08, 44%, 38% and 2.2 for ultra-peripheral areas, respectively). Rural municipalities exhibit a higher aging index (2.2) compared to towns (1.4) and cities (1.2), along with larger values of low educational index (0.61 vs. 0.58 in towns and 0.55 in cities) and dependency ratios (0.58 vs. 0.52 in towns and 0.50 in cities). Conversely, these rural areas show smaller values of high educational index (0.09 compared to 0.11 in towns and 0.15 in cities), employment rates (0.48 vs. 0.52 in towns and cities), and crowding index (2.3 compared to 2.5 in towns and 2.6 in cities).

Social vulnerability is often expressed through a composite index known as the Social Vulnerability Index (SoVI), which aggregates different metrics affecting it (e.g., Cutter et al., 2003; Frigerio et al., 2018). Using a unique index to represent social vulnerability provides a comprehensive and easily interpretable measure that encapsulates multiple dimensions of vulnerability, facilitating communication and policymaking. In this study the individual indicators affecting social vulnerability are considered in the cluster analysis, while the aggregated SoVI index is used to provide a synthetic description. This approach allows for a more nuanced understanding of the different dimensions of vulnerability. By analysing each indicator separately, cluster analysis can capture the unique contributions and relationships between factors like income, education, health, and housing quality, which may be masked in a single composite index. Additionally, considering individual indicators enables the identification of distinct patterns or subgroups within the data, leading to more effective archetype identification. In contrast, an aggregated index may oversimplify these dynamics and overlook important variations across clusters.

## 3    Cluster analysis

To identify archetypes by grouping entities based on shared characteristics, clustering analysis is widely used. Clustering refers to unsupervised learning techniques used to find subgroups (or clusters) within a data set by organizing elements according to their similarities. This method is designed to group together observations that are highly similar, while separating those that differ into distinct clusters. Unlike supervised classification algorithms, which rely on labelled training data to categorize new information into predefined classes, clustering uncovers natural structures in the data by analysing similarities between data points without the need for predefined labels.

In this study clustering is adopted to group human settlements based on their potential exposure and vulnerability-related factors to define risk-oriented urban and rural settlements archetypes, proposing an application for Italian municipalities as a case. We conducted a two-step clustering approach. A first cluster analysis is performed with a sub-set of attributes, specifically demographic and geographic parameters representative of physical and institutional vulnerability dimensions. The aim is to allow a first broad definition of archetypes, simple and highly interpretable. In the second step, a nested clustering approach is applied to further differentiate sub-clusters based on socio-economic attributes.

Both hierarchical and partitioning clustering techniques are employed in each step to enable comparison of clustering outputs and to identify nuanced patterns that may not be captured by a single method. The adoption of two different clustering techniques serves to enhance the robustness, reliability, and interpretability of the archetypes identified in this

study. Each method has distinct strengths and analytical advantages, which, when combined, allow for a more comprehensive exploration of patterns in the data. For instance, hierarchical clustering is particularly useful for exploring data structures without the need to predefine the number of clusters. It produces a dendrogram that visually represents nested groupings and their relationships, offering insights into how clusters evolve as dissimilarity thresholds change. This is especially valuable for understanding the hierarchical nature of urban/rural systems and guiding the selection of an appropriate number of clusters. On the other hand, partitioning clustering requires the number of clusters to be predefined, but it typically performs better with larger datasets, producing compact, well-separated clusters when appropriately parameterized. It is computationally efficient and more suitable for refining clusters, especially when working with both categorical and numerical data types. Using both techniques enables cross-validation of clustering outputs, ensuring consistency and increasing confidence in the identified archetypes. Discrepancies between methods can highlight ambiguous or transitional settlement types, while convergences confirm stable, well-defined clusters.

More detailed information of algorithms used are reported in section 3.2 and 3.3. Results of first and second level clustering analysis are presented in section 4 and 5, respectively. Rural and urban archetypes (presented in section 6) are defined based on the results of the most effective algorithm, selected according to widely used clustering performance metrics (see section 4.2).

## 3.1    Data pre-processing

Data preprocessing is crucial for enhancing the quality of clustering. Specifically, we performed: (i) outlier detection; (ii) correlation analysis – to eliminate measurement redundancy; (iii) normalization of numerical data values.

The detection of outliers is necessary to ensure high quality of clustering. An outlier is an object in a data set that deviates significantly from the remaining data, Outlier detection is essential for ensuring high-quality clustering, as extreme values can distort normalization and affect cluster formation (Nowak-Brzezińska & Gaibei, 2022; OECD, 2008).Outliers are identified using the interquartile range (IQR) method, where data points beyond 1.5 times the IQR from the quartiles are considered outliers. In this study, residential population had the highest number of outliers. Instead of removing them, which would compromise the analysis, the population variable was transformed into categorical classes (i.e. the population classes presented in section 3.4) to reduce its impact on clustering.

For the correlation analysis, the Pearson correlation coefficient ($r$) is used. This coefficient is a statistical measure used to assess the strength and direction of the linear relationship between two continuous variables (Cohen, 2013). It is one of the most used methods for correlation analysis. While it does not inherently assume normality, research indicates that Pearson's correlation is relatively robust to violations of normality, especially when sample sizes are large (Bishara & Hittner, 2017). The value of $r$ ranges between -1 and 1, with values higher than 0 that indicate a positive correlation and values lower than 0 a negative correlation. Values close to 0 indicate no linear correlation between variables (Cohen, 2013). Figure 3 shows the correlation matrix obtained for the 14 numerical variables presented in section 2.5. The analysis shows that there is a very strong correlation ($| r | > 0.8$) between age indicators, namely: proportion of population under 15 aged and over 65 aged ($r = -0.84$); aging index and proportion of people aged 65 and above ($r = +0.81$); dependency ratio and proportion of population over 65 ($r = 0.91$). Strong correlation ($| r | > 0.5$) is also observed for: aging index and dependency ratio to proportion of people under 15 ($r$ equal to -076 and –0.55, respectively); dependency ratio and aging index ( $r = +0.73$); crowding index and dependency ratio ( $r = -0.6$); crowding index and proportion of families with 5 or more components ($r = +0.78$), high and low educational index ($r = -0.78$), proportion of commuters and proportion of employed ($r = +0.61$); proportion of employed and proportion of over 65 aged ($r = -0.58$) and proportion of under 15 aged ( $r = +0.54$).

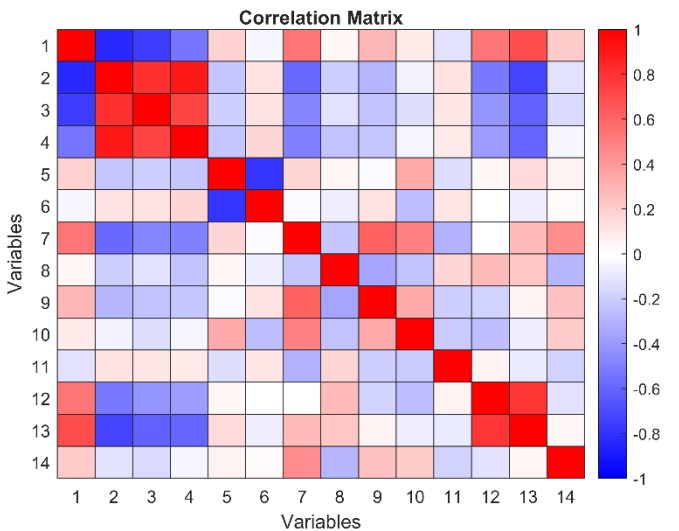

**Figure 3 – Correlation matrix for numerical variables considered.**

Based on the analysis results, the number of numerical attributes used for clustering was reduced from 14 to the following 8: aging index, low educational index, proportion of unemployed, proportion of commuters, proportion of female employed, proportion of buildings in poor condition, crowding index and proportion of foreign resident. The attributes selected for the clustering are thus divided into 4 categorical (i.e., degree of urbanisation, population class, degree of urban centeredness and altimetric zone) and 8 numerical, listed in table 3.

**Table 2 – Variable used in cluster analysis.**

| Variable | Type | Vulnerability dimension |
|---|---|---|
| Degree of urbanisation | Categorical | Physical |
| Degree of urban centeredness | Categorical | Institutional |
| Residential Population class | Categorical | Physical |
| Altimetric zone | Categorical | Institutional |
| Aging index | Numerical | Social |
| Low educational index | Numerical | Social |
| Unemployed | Numerical | Economic |
| Commuting rate | Numerical | Social |
| Female employed | Numerical | Economic |
| Quality of buildings | Numerical | Social |
| Crowding index | Numerical | Social |
| Foreign resident | Numerical | Social |

Finally, numerical data are normalized to enhance the quality of clustering by ensuring that all features contribute equally to the analysis, regardless of their original scales. Without normalization, features with larger ranges could dominate the clustering process, leading to biased results (Usman & Stores, 2020). As normalization method, the empirical cumulative distribution function (ECDF) is adopted. The empirical CDF approach ranks the data points by their cumulative probability, effectively distributing them between 0 and 1 based on their relative positions within the dataset. Compared to other normalization methods (e.g., min-max normalization, z-score), ECDF normalization offers several advantages: it effectively processes non-normally distributed data, minimizes the impact of outliers, and provides a clear, intuitive framework for interpreting data rankings relative to the overall distribution (Hoffman et al., 2017).

### 3.2 Hierarchical clustering

Hierarchical clustering organizes data into tree-structured clusters through either an agglomerative or divisive process (Han et al., 2011). In agglomerative clustering, each object is initially assigned to an individual cluster (that is, if there are $n$ objects, the process will start with $n$ clusters). Initial clusters are gradually merged into larger clusters based on their similarity (or dissimilarity), until a hierarchy of clusters is built and only one cluster remains, which contains all data points. The selection of an appropriate distance or dissimilarity measure crucially affects the clustering solution and depends on the nature of the considered variables. Most distance measures concern the analysis of either continuous only or categorical only data (e.g., Euclidean distance). The Gower distance (Gower, 1971) is a flexible dissimilarity measure that can work both with numerical and categorical variables. For numerical variables, the Gower distance uses the

normalized absolute difference. If $x_{il}$ and $x_{jl}$ are the values of the *l-th* numerical attribute for objects $i$ and $j$, the distance between the two objects for attribute $l$ is calculated as:

$$d_{ij}^l = \frac{|x_{il} - x_{jl}|}{\max(x_l) - \min(x_l)} \tag{1}$$

For categorical variables, the Gower distance assigns a value of 0 if the values are the same and 1 if they are different:

$$d_{ij}^l = \begin{cases} 0 & if\ x_{il} = x_{jl} \\ 1 & if\ x_{il} \neq x_{jl} \end{cases} \tag{2}$$

The Gower distance for a pair of objects $i$ and $j$ is calculated as the average value of the individual attribute distances, according to Eq. (3):

$$D_{ij} = \frac{\sum_{l=1}^{p} w_l d_{ij}^l}{\sum_{l=1}^{p} w_l} \tag{3}$$

Where $p$ is the number of attributes, $d_{ij}^l$ is the individual attribute distance and $w_l$ is the weight for the *l-th* attribute, set to 1.

The main steps of agglomerative hierarchical clustering process can be outlined as follows:

1) For each pair of data points $i$ and $j$ in the dataset, the Gower distance is calculated. The result is an $n \times n$ distance matrix where each entry $(i, j)$ represents the Gower distance between objects $i$ and $j$ across all attributes. This means that the values on the diagonal of this matrix will be equal to zero, since the distance between object $i$ and itself is zero. It is important to note that the Gower distance already normalizes numerical variables, making additional normalization unnecessary.

2) Next, the two clusters with the smallest Gower distance are merged, reducing the total number of clusters by one.

3) The distance matrix is then updated to reflect the distance between the newly formed cluster and all other clusters. The recalculation of distances depends on the chosen linkage criterion (e.g., single, complete, average, or ward linkage). In this study, complete linkage is employed, where all pairwise dissimilarities between observations in cluster A and cluster B are computed, and the largest of these dissimilarities is recorded.

4) Repeat the merging process iteratively, continuing to merge the closest clusters and updating the distance matrix until all objects are grouped into a single cluster.

### 3.2.1 Optimal number of clusters

Throughout the merging process, a dendrogram is constructed—a tree-like diagram that visually represents the order and levels at which clusters are merged. The height of each node in the dendrogram corresponds to the Gower distance at which the clusters were merged. The dendrogram can be analysed to determine the optimal number of clusters, either visually by identifying the largest vertical distance (gap) between merges, known as the "cut" point (James et al., 2017), or by evaluating the inconsistency coefficient (Martin et al., 2022). The inconsistency coefficient measures the similarity of clusters connected by each link, comparing its length with the average length of other links at the same level of the dendrogram (Jatain et al., 2013). A higher coefficient indicates less similarity between clusters. The relationship between the inconsistency coefficient and the number of clusters indicates that a lower number of clusters corresponds to higher inconsistency, which suggests better clustering since distinct clusters tend to have high inconsistency. However, fewer clusters often lead to greater within-cluster variance. To strike a balance between distinct clusters and minimizing within-cluster variance, the variability of observations within each cluster is evaluated, and its trend is analysed as the number of clusters increases.

To evaluate the variability of the observations within each cluster, a coefficient representing the within cluster distance (WCD) is calculated as sum of the average values of distances between data points in a single cluster. Specifically, for the *l-th* numerical attribute $WCD_l$ is calculated by taking the average of the squares of the differences between each pair of values $i$ and $j$ in the cluster $C_k$ (Gordon, 1986):

$$WCD_l(C_k) = \frac{\sum_{i=1}^{n} \sum_{j=1}^{n} (x_{il} - x_{jl})^2}{n^2 - n} \tag{4}$$

Where $x_1, x_2, ..., x_n$ are $n$ observations within the *k-th* cluster on a quantitative variable, $x$.

For the *l-th* categorical attribute, the coefficient of unlikeability proposed by Perry & Kader (2005) is utilized as measure of $WCD_k$:

$$WCD_l(C_k) = \frac{\sum_{i \neq j} c(x_{il}, x_{jl})}{n^2 - n} \tag{5}$$

Where:

$$c(x_i, x_j) = \begin{cases} 1 & if \ x_{il} \neq x_{jl} \\ 0 & if \ x_{il} = x_{jl} \end{cases} \tag{6}$$

And $x_1, x_2, ..., x_n$ are $n$ observations within the *k-th* cluster on a categorical variable, $x$.

The final value of WCD for the cluster $C_k$ is given by the average values across all $p$ attributes:

$$WCD(C_k) = \frac{\sum_{l=1}^{p} WCD_l(C_k)}{p} \tag{7}$$

While the overall value of WCD for the clustering – which accounts for all $m$ clusters - can be defined as the average value:

$$WCD = \frac{\sum_{k=1}^{m} WCD(C_k)}{m} \tag{8}$$

The evaluation of variation of WCD with the increasing number of clusters together with the trend of inconsistency coefficient allows the definition of the best number of clusters for the specific case analyses.

### 3.3 Partitioning clustering

Partitioning clustering is a method that divides a dataset into a predefined number of clusters by assigning each data point to a single cluster based on similarity. The most used partitioning clustering technique is the k-means algorithm (MacQueen, 1967). To perform k-means clustering, the number $k$ of clusters must be predefined, and $k$ objects, representing the initial cluster centroids, are arbitrarily chosen. The remaining objects are then iteratively assigned to these clusters in a way that minimizes the distances of points to their respective centroids, thereby minimizing the within-cluster variance. The position of each centroid is updated each iteration by the mean value of the objects in a cluster. One of the main drawbacks of k-means algorithm is that it only works on numeric values, prohibiting its use to cluster data containing categorical values. The k-modes algorithm is an extension of k-means algorithm that employs a simple matching dissimilarity measure to handle categorical data, replacing cluster means with modes and using a frequency-based approach to update these modes during the clustering process (Huang, 1998). These modifications enable the k-modes algorithm to cluster categorical data in a manner similar to k-means. The k-prototypes algorithm combines elements of the K-means algorithm and the K-modes algorithm, allowing for the clustering of objects characterized by both numeric and categorical attributes (Huang, 1998). Like the k-means algorithm, this technique requires the user to set the number of clusters ($k$), while initial cluster centroids are chosen arbitrarily. Observations are iteratively assigned to the closest centroid in a way that minimizes within-cluster variance. To define the closeness between two objects, this method applies Euclidean distance to numeric attributes and uses a distance function simple matching dissimilarity ($\delta$=0 if the values match, $\delta$=1 if they do not) for categorical attributes. Thus, dissimilarity measure for a data point $i$ and centroid $j$ can be calculated as:

$$D(i,j) = \sum_{numerical} (x_{il} - \mu_{kl})^2 + \gamma \sum_{categorical} \delta(x_{il}, \mu_{kl}) \tag{9}$$

Where $x_{il}$ is the value of the *l-th* attribute of the $i$-th data point, $\mu_{kl}$ is the relative value for the *k-th* cluster centroid, and $\gamma$ is a weighting factor to balance numerical and categorical distances (a value of 1 is adopted for $\gamma$). The centroids are updated after each iteration by taking the average values of numerical variables of the objects within a cluster and evaluating the modes for categorical attributes, i.e., the category with the highest frequency.

As one of the main drawbacks of this clustering techniques is that the clustering is very sensitive to the selection of initial centroid, the random selection of initial centroids and the clustering are repeated $t$ times ($t$=10) and each iteration $t$ the performance of the clustering algorithm is evaluated by calculating WCD. Among $t$ different clustering obtained, the best clustering is determined based on WCD value (i.e., the clustering providing the lower WCD value).

## 4    First level analysis: clustering based on physical and institutional vulnerability parameters

A preliminary cluster analysis is conducted using only demographic and geographic attributes representative of physical and institutional vulnerability dimensions, namely: degree of urbanisation, residential population, centeredness degree and altimetric zone. It is important to note that all these attributes are categorical (see Table 2). Consequently, hierarchical clustering is performed using Eq. (2) and (3), while partitioning clustering is applied using the dissimilarity measures for categorical variables presented in Eq. (9).

The optimal number of clusters is determined following the procedure outlined in Section 4.2.1. The inconsistency coefficient reaches its highest values (>4) when considering between 4 and 12 clusters. Conversely, the within-cluster dispersion (WCD) exhibits an inverse relationship with the inconsistency coefficient: as the number of clusters increases, WCD decreases. Specifically, WCD declines from 1.26 with 4 clusters to 0.70 with 10 clusters, remaining constant at 0.70 between 10 and 12 clusters. To achieve a balanced trade-off between the inconsistency coefficient and WCD, 10 clusters are selected as the optimal number for this dataset. Since partitioning clustering requires a predefined number of clusters, the optimal number identified through this methodology is also adopted for the partitioning clustering approach.

### 4.1    Results of hierarchical cluster analysis

Figure 4 shows the representativeness of clusters in terms of attributes considered. Clusters 1, 2, 3 and 6 represent rural municipalities with low (1, 2 and 3) and medium (6) population. Among them, cluster 1 identifies hubs (more specifically intermunicipal hubs), cluster 2 peripheral areas, cluster 3 peri-urban areas while cluster 6 includes both peripheral and peri-urban municipalities – with a very small portion of intermunicipal hubs.

Clusters 4, 5 and 7 identifies medium-density towns and suburbs with medium (4), medium-low (7) and low population (5). Cluster 4 is represented by hubs (specifically, intermunicipal hubs), cluster 7 by peripheral areas while cluster 5 in majority by peri-urban areas. Clusters 8 and 9 include high density cities, that are medium (8) and high (9) populated areas. Cluster 9 includes all the major Italian cities with more than 250'000 inhabitants (e.g., Rome, Milan and Naples), while cluster 8 includes cities located in peri-urban areas. Finally, cluster 10 includes all medium-populated municipalities not included in the other clusters, most of the medium-densely populated and located in peri-urban and peripheral areas.

Regarding the altimetric zone, most peripheral areas (e.g., clusters 2 and 7) are located in hilly and mountainous regions, whereas densely populated cities (e.g., clusters 8 and 9) are primarily situated in lowland areas. Cluster 4 distinctly represents towns and suburbs in hilly regions, while Cluster 6 includes rural municipalities in mountainous areas. Cluster 1 groups intermunicipal hubs found in both hilly and mountainous areas. However, the classification of municipalities in clusters 3, 5, and 10 is less clearly defined.

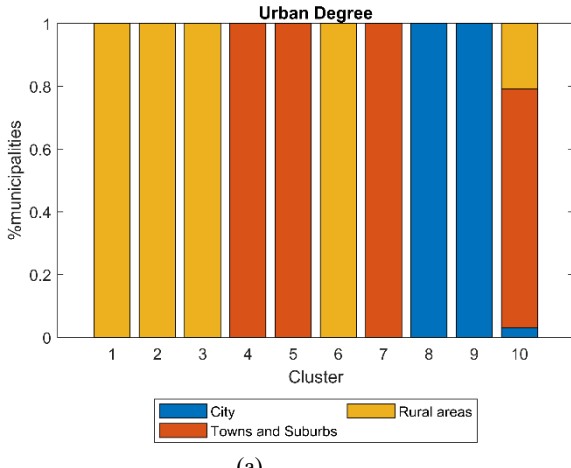

(a)

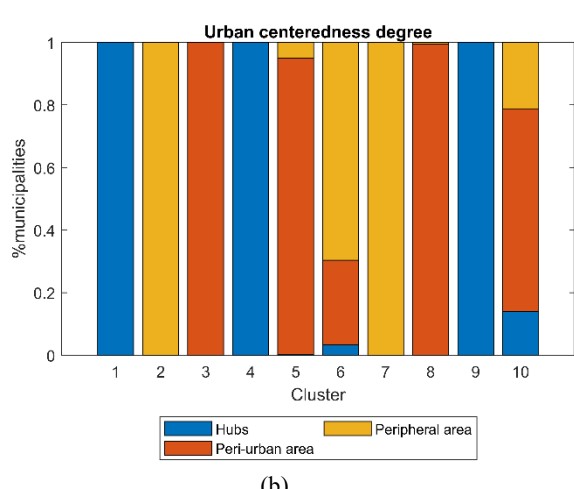

(b)

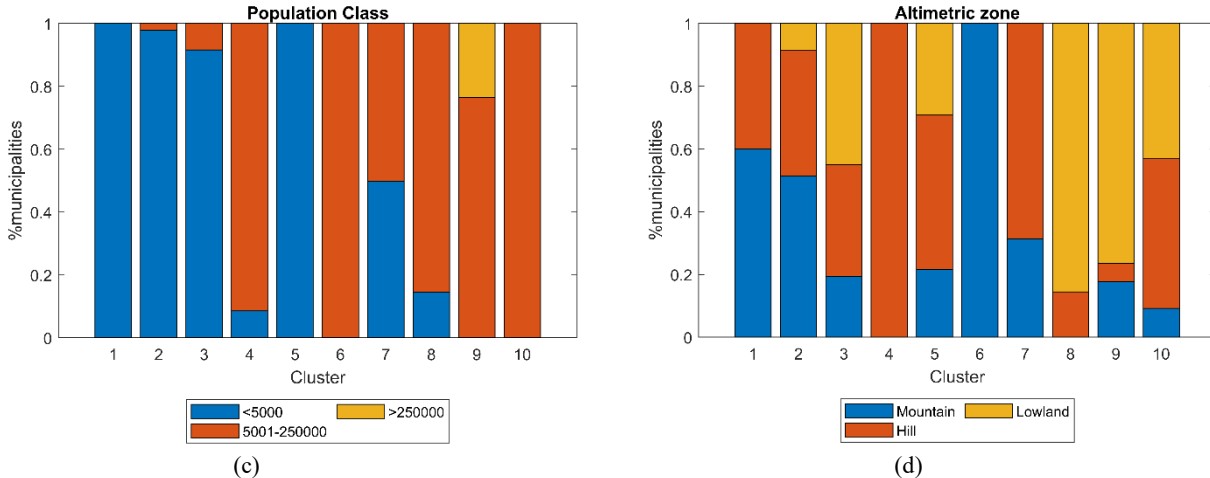

(c)              (d)

**Figure 4 - Representativeness of clusters resulting from hierarchical clustering in terms of degree of urbanisation (a), degree of urban centeredness (b), population class (c) and altimetric zone (d).**

The attributes' importance for clustering is evaluated adopting simplified procedure proposed in Fraiman et al. (Fraiman et al., 2008). The methodology is based on an iterative removal of variables, to assess their contribution to clustering based on the performance metric selected. In other words, variables are removed one at a time and the impact of the removal on the overall model is measured. The greater the impact of removing a variable, the more important it is considered. In this study we consider WCD as the performance metric. Figure 5a shows that the most important attribute for the clustering is the degree of urbanisation, followed by the population class and the centeredness degree, while the less important attribute is the altimetric zone.

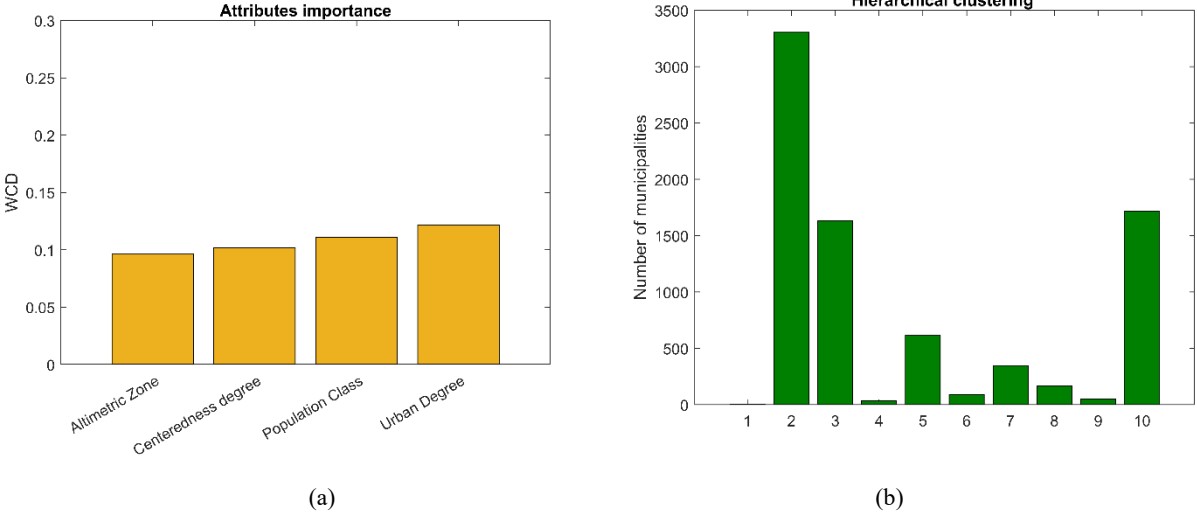

(a)              (b)

**Figure 5 – Attribute importance in terms of variation of WCD (a); number of Italian municipalities belonging to each cluster (b).**

A significant number of municipalities are classified within rural clusters, with Cluster 2 encompassing 3,305 municipalities and Cluster 3 including 1,632 municipalities (Figure 5b). This aligns with the fact that 68% of Italian municipalities are categorized as rural areas (see also Section 3.4). The least populated cluster is Cluster 1, which contains only 5 municipalities, followed by Cluster 4 (35 municipalities), Cluster 9 (51 municipalities), and Cluster 6 (89 municipalities). Meanwhile, Clusters 8, 7, 5, and 10 include 166, 347, 615, and 1,715 municipalities, respectively.

## 4.2  Results of partitioning cluster analysis

Results of partitioning clustering are presented in Figure 6, highlighting the representativeness of different attributes. Clusters 3 and 10 represent low populated peripheral municipalities in rural areas, specifically located in mountainous (3) and hilly (10) regions. Clusters 1, 6 and 8 also include low populated municipalities in rural areas but classified as peri-urban, located in hilly (1), lowland (6) and mountainous areas (8). Clusters 4 and 9 identify low populated suburban

municipalities in peri-urban areas, with cluster 4 representing those in mountainous and hilly regions, and cluster 9 those in lowland areas. Clusters 2 and 5 characterize medium-low (5) and medium populated (2) towns in peripheral and peri-urban areas. Municipalities in cluster 5 are predominantly located in mountainous regions, while those in cluster 2 are mainly found in hill and lowland areas. Finally, cluster 7 includes medium to high populated cities, encompassing both hubs and peri-urban municipalities, primarily located in lowland regions.

The attribute importance analysis (Figure 7a) indicates that degree of urbanisation is the most influential factor in accurately distinguishing clusters, followed by altimetric zone, centeredness degree, and population classification.

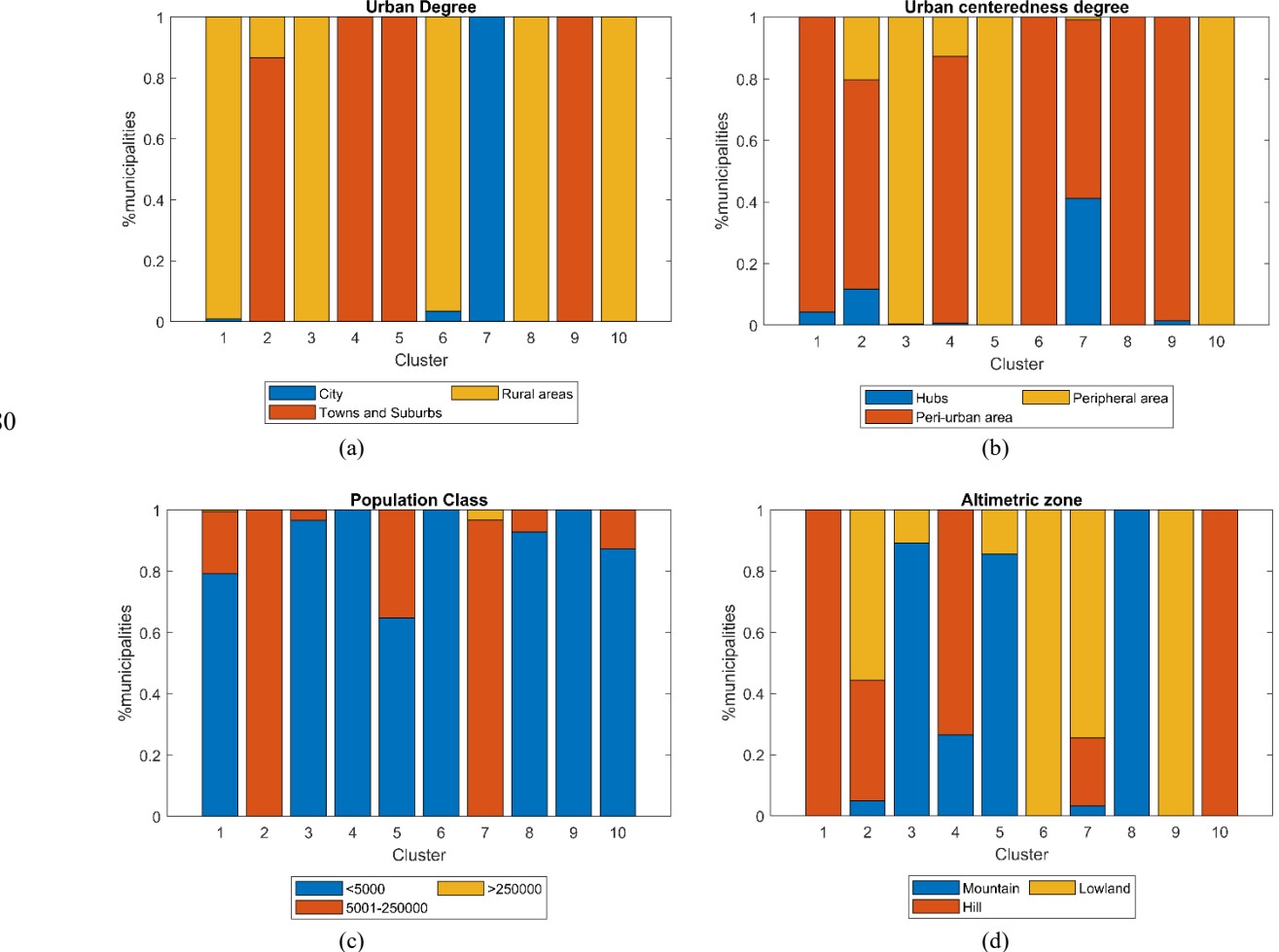

**Figure 6 - Representativeness of clusters resulting from partitioning clustering in terms of degree of urbanisation (a), degree of urban centeredness (b), population class (c) and altimetric zone (d).**

The distribution of municipalities across clusters varies significantly (Figure 7b). Cluster 3 is the largest, comprising 1,976 municipalities, followed by Cluster 2 with 1,655 municipalities and Cluster 10 with 1,520 municipalities. Cluster 1 includes 740 municipalities, while Cluster 6 and Cluster 8 contain 618 and 341 municipalities, respectively. Cluster 4 and Cluster 5 represent smaller groups, with 503 and 216 municipalities. The smallest clusters are Cluster 9 with 148 municipalities and Cluster 7 with 243 municipalities, indicating distinct groupings within the dataset.

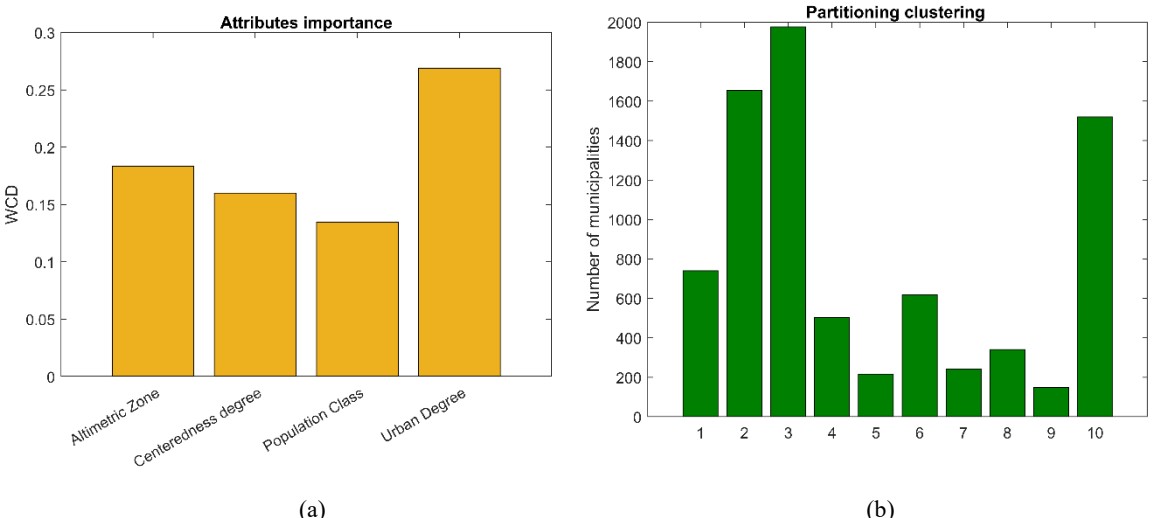

|         |         |
|:-------:|:-------:|
|   (a)   |   (b)   |

**Figure 7 - Attribute importance in terms of variation of WCD (a); number of Italian municipalities belonging to each cluster (b).**

### 4.3 Comparison of clustering algorithms

In order to evaluate quality of clustering techniques, measure of intra-cluster distance (i.e., WCD presented in section 4.2.1) as well as inter-clusters distance are investigated. Specifically, the coefficient WCD is adopted to evaluate the performance of the clustering for each single attribute. The value of WCD$_l$ for the attribute *l-th* across all clusters is calculated as follows:

$$WCD_l = \frac{\sum_{k=1}^{m} WCD_l(C_k)}{m} \tag{10}$$

Where $WCD_l(C_k)$ is the value of WCD for the *l-th* attribute and the *k-th* cluster, calculated according to Eq. (4) and Eq. (5), and *m* is the total number of clusters. The lower the WCD$_l$ value, the better the performance of the algorithm with respect to the considered attribute.

Inter-cluster distance (ICD) measures the separation between clusters in a clustering solution and is useful for evaluating how distinct the clusters are. To calculate inter-cluster distance, we adopt Centroid-to-Centroid Distance:

$$ICD_l = \frac{\sum_{i \neq j} \|\mu_{i,l} - \mu_{j,l}\|}{m} \tag{11}$$

Where $\mu_{i,l}$ and $\mu_{j,l}$ are the centroids values (i.e., the mode of the objects within a cluster) of clusters *i* and *j* for the *l-th* attribute. Unlike WCD, a higher ICD indicates better algorithm performance, as it reflects greater differentiation between clusters.

From Figure 8, it can be observed that the hierarchical clustering algorithm achieves better WCD performance for the centeredness degree attribute. However, it performs worse than partitioning clustering for the population class attribute and significantly worse for the altimetric zone attribute. Regarding degree of urbanisation, both clustering methods exhibit high and comparable WCD performance. Overall, considering the average WCD across all attributes, the hierarchical algorithm shows a higher WCD (0.18) compared to the partitioning algorithm (0.13). Despite this difference, both clustering approaches demonstrate relatively good performance. In terms of ICD, hierarchical clustering outperforms partitioning clustering across all attributes, except for the altimetric zone, where both methods yield the same ICD value. Overall, hierarchical clustering demonstrates superior performance in terms of ICD, with a value of 0.62, compared to 0.49 for partitioning clustering.

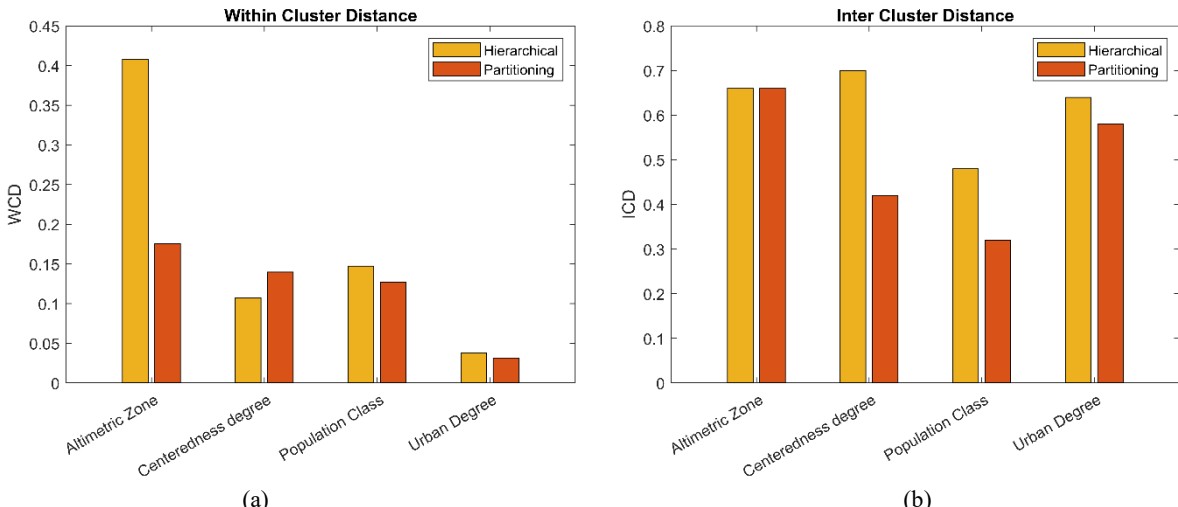

(a)                                         (b)

**Figure 8 – Comparison of clustering algorithms in terms of WCD$_I$.(a) and ICD$_I$ (b).**

While WCD is slightly higher in hierarchical clustering compared to partitioning clustering, the overall clustering quality remains good in both methods. However, the significantly higher ICD (0.62 vs. 0.49) for hierarchical clustering suggests that it produces more distinct and well-separated clusters, making it the preferable choice for this analysis. Therefore, the results of hierarchical clustering are used for the initial broad definition of archetypes (presented in Section 5) and serve as the input for the second step of the analysis, namely the nested clustering, which is discussed in the following section.

**5     Second level analysis: nested clustering based on socio-economic parameters**

Nested clustering identifies clusters within clusters, unveiling data structures at multiple levels of granularity. This method is particularly valuable for detecting complex patterns in data, providing a more detailed understanding of the underlying relationships. In this study, for each first-level cluster (or broad archetype) identified in the previous section, we analyse nested clusters to capture the heterogeneity of urban/rural settlements in terms of socio-economic vulnerability. To achieve
this, we consider eight socio-economic attributes—aging index, low educational index, proportion of unemployed individuals, proportion of commuters, proportion of female employees, proportion of buildings in poor condition, crowding index, and proportion of foreign residents—selected based on the correlation analysis results presented in Section 4.1.

Both partitioning and hierarchical clustering algorithms are applied within each cluster to further refine the sub-groups.
The optimal number of sub-clusters for each cluster is determined using WCD and the inconsistency coefficient, as detailed in Section 4.2.1. Based on the clustering performance metrics (i.e., WCD and ICD), partitioning clustering proves to be the most suitable approach for this second-level clustering. As results, we identified 3 sub-clusters for cluster 2 and cluster 3, 2 sub-clusters for cluster 4, 8, 9 and 10 and no sub-clusters for cluster 1, 5, 6 and 7 due to homogeneity of socio-economic data within the cluster. The list of clusters and sub-clusters with the average values of numerical attributes for
each of the sub-clusters identified is reported in Table 4.

While cluster names are derived from the geographical and demographic characteristics analysed (e.g., peri-urban settlements, peripheral rural areas), sub-cluster names are assigned based on the mean value of the Social Vulnerability Index (SoVI) within each sub-cluster, which reflects demographic and socio-economic conditions of the settlements. Additionally, if necessary, sub-cluster names may also incorporate the specific social vulnerability factors that contribute
most significantly to the SoVI value. For example, both sub-clusters 2a and 2b exhibit high social vulnerability; however, sub-cluster 2b has the highest aging index among all sub-clusters, leading to its classification as "*aged communities with high social vulnerability*." Similarly, sub-clusters 3a ("*aged communities with high social vulnerability*") and 3b ("*high household density settlements with high social vulnerability*") both exhibit high SoVI values, but the former is characterized by a high aging index, while the latter has a high crowding index. Understanding the influence of individual
socio-economic indicators within each archetype can support the prioritization and tailoring of risk mitigation strategies and resilience policies. The SoVI indicator is calculated following the procedure proposed by Frigerio et al.(2018), utilizing the same socio-economic variables adopted in this study for clustering. The criteria used to identify the different socio-economic condition categories is based on SoVI values and specifically: a value lower than 1 corresponds to low social vulnerability (dark green in Table 4), value between 1 and 1.20 to moderate social vulnerability (light green in

Table 4), values between 1.20 and 1.40 to intermediate social vulnerability (yellow in Table 4), values between 1.40 and 1.60 to high social vulnerability (light red in Table 4), values higher than 1.60 to very high social vulnerability (dark orange in Table 4). The average values of individual variables for each sub-cluster are provided in Table 4.

**Table 3 – Cluster and sub-clusters identified with relative average value of numerical attributes within the cluster or sub-cluster.**

| Cluster | Sub-cluster | Aging Index | Low Educational Index | Unemployed | Comm uters | Female employed | Building poor state | Crowding index | Foreigns | SoVI |
|---|---|---|---|---|---|---|---|---|---|---|
| 1. Low populated intermunicipal hubs in rural areas | - | 1.69 | 0.54 | 0.03 | 0.23 | 0.44 | 0.14 | 2.44 | 0.09 | 1.31 |
| 2. Low populated peripherical rural areas | a. High household density settlements with high social vulnerability | 2.04 | 0.61 | 0.04 | 0.22 | 0.39 | 0.19 | 2.40 | 0.04 | 1.50 |
| | b. Aged communities with high social vulnerability | 3.70 | 0.68 | 0.02 | 0.28 | 0.38 | 0.23 | 2.03 | 0.05 | 1.54 |
| | c. Aged communities with moderate social vulnerability | 3.31 | 0.59 | 0.03 | 0.19 | 0.43 | 0.19 | 2.12 | 0.04 | 1.15 |
| 3. Medium-Low populated peri-urban settlements in rural areas | a. Aged communities with high social vulnerability | 2.38 | 0.61 | 0.03 | 0.34 | 0.44 | 0.22 | 2.21 | 0.05 | 1.46 |
| | b. High household density settlements with high social vulnerability | 1.51 | 0.56 | 0.05 | 0.26 | 0.39 | 0.21 | 2.59 | 0.03 | 1.56 |
| | c. Settlements with high social vulnerability | 1.78 | 0.62 | 0.03 | 0.34 | 0.42 | 0.12 | 2.47 | 0.08 | 1.54 |
| 4. Medium-populated intermunicipal hubs | a. Settlements with intermediate social vulnerability | 1.77 | 0.54 | 0.03 | 0.28 | 0.45 | 0.10 | 2.40 | 0.08 | 1.22 |
| | b. Settlements with high social vulnerbaility | 1.22 | 0.55 | 0.05 | 0.15 | 0.40 | 0.18 | 2.74 | 0.03 | 1.40 |
| 5. Low Populated peri-urban suburbs | - | 1.38 | 0.58 | 0.04 | 0.34 | 0.41 | 0.15 | 2.55 | 0.05 | 1.53 |
| 6. Medium-populated peri-urban and peripheric suburbs | - | 1.71 | 0.54 | 0.03 | 0.14 | 0.41 | 0.19 | 2.50 | 0.04 | 1.17 |
| 7. Peripheric suburbs medium-low populated | - | 1.43 | 0.59 | 0.04 | 0.24 | 0.40 | 0.19 | 2.50 | 0.06 | 1.51 |
| 8. Peri-urban cities | a. Settlements with very high social vulnerability | 0.73 | 0.63 | 0.06 | 0.22 | 0.36 | 0.26 | 3.08 | 0.02 | 2.01 |
| | b. Settlements with intermediate social vulnerability | 1.17 | 0.55 | 0.03 | 0.40 | 0.45 | 0.11 | 2.44 | 0.07 | 1.38 |
| 9. Major urban hubs | a. Settlements with low social vulnerability | 1.95 | 0.47 | 0.03 | 0.11 | 0.47 | 0.10 | 2.23 | 0.11 | 0.84 |
| | b. Settlements with intermediate social vulnerability | 1.50 | 0.51 | 0.05 | 0.08 | 0.42 | 0.24 | 2.60 | 0.03 | 1.20 |
| 10. Medium-populated towns | a. Settlements with high social vulnerbaility | 1.15 | 0.57 | 0.05 | 0.18 | 0.38 | 0.22 | 2.71 | 0.03 | 1.56 |
| | b. Settlements with intermediate social vulnerability | 1.54 | 0.57 | 0.03 | 0.28 | 0.43 | 0.13 | 2.48 | 0.08 | 1.39 |

## 6    Urban and rural settlements archetypes in Italy

The first-level clustering provides a "broad" definition of archetypes, considering only geographic and demographic attributes. Clusters 1, 2 and 3 ("*Low populated intermunicipal hubs in rural areas*", "*Low populated peripherical rural areas*" and "*Low populated peri-urban settlements in rural areas*", see table 4) represent archetypes of low populated, rural urban settlements in peripherical (2) and peri-urban (3) areas or close to urban hubs (1). Cluster 5 ("*Low Populated peri-urban suburbs*") also represents archetypes for low populated settlements but characterized by higher population density. Clusters 4, 6, 7 and 10 ("*Medium-populated intermunicipal hubs*", "*Medium-populated peri-urban and peripheric suburbs*", "*Peripheric suburbs medium-low populated*" and "*Medium-populated towns*") are archetypes for medium-populated peri-urban and peripheric suburbs (6, 7, 10) and towns that are intermunicipal hubs (4). Clusters 8 and 9 ("*Peri-urban cities*", "*Major urban hubs*") represent archetypes for densely populated peri-urban settlements (8) and hubs (9). These broad archetypes are mapped in Figure 9. Notably, altimetric classification is not included in the archetype definition, as it exhibits high within-cluster variance and is the least significant attribute, making its contribution negligible in defining urban and rural archetypes.

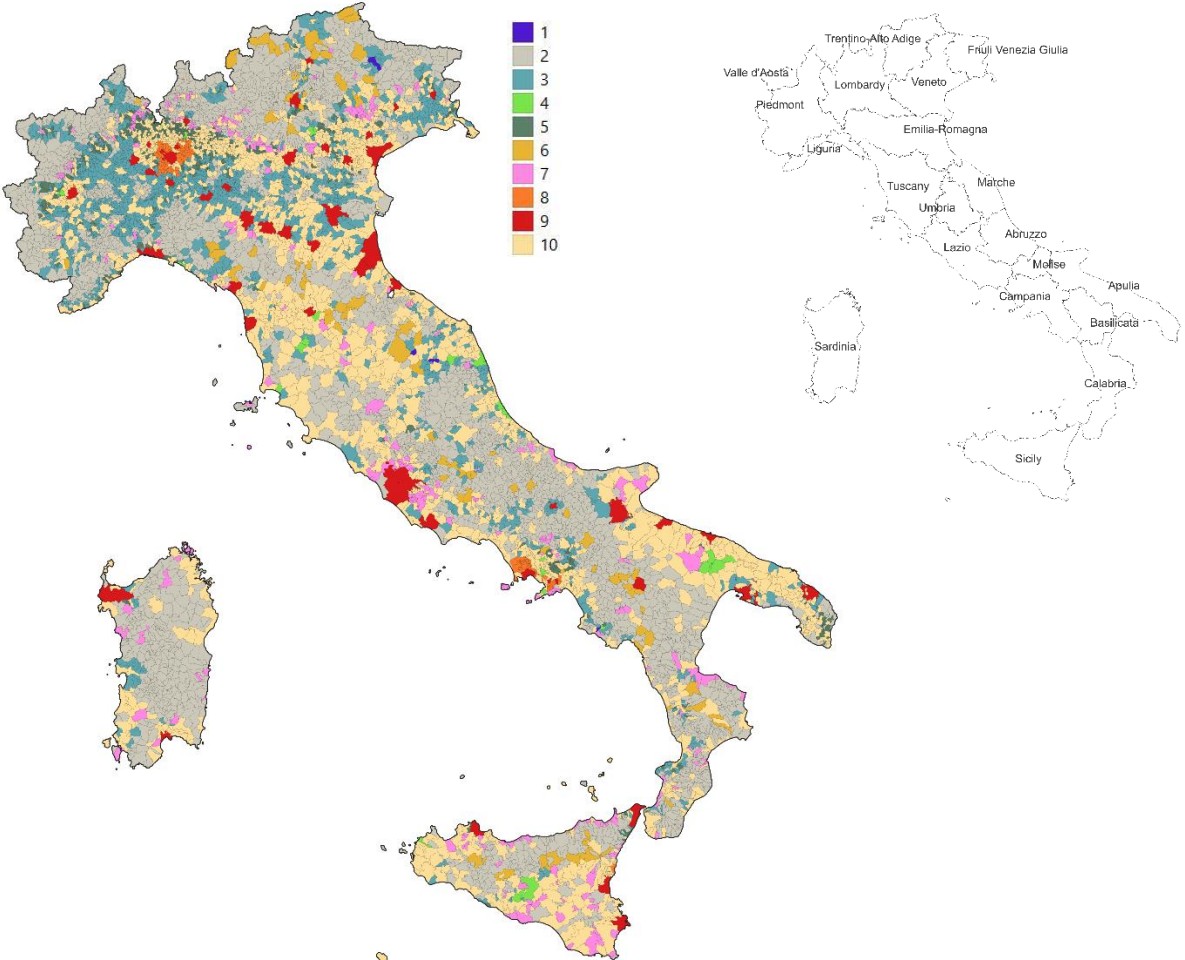

**Figure 9 – Broad urban and rural archetypes across Italian territory. On the right the map with the Italian regions is reported.**

The proposed archetypes are obtained considering both first-level clusters and sub-clusters. Specifically, 18 urban and rural archetypes are defined (listed in Table 5), characterized by geographic, demographic, and socio-economic features. Each archetype is identified by an alphanumeric code and a designation, derived from the combination of the codes and names of the clusters and sub-clusters from which they are obtained.

**Table 4 – Urban and rural archetypes identified. The number of municipalities belonging to each archetype and the related population share are provided in the last columns.**

| | Archetypes | n° municipalities | Population share (%) |
|---|---|---|---|
| 1 | Low populated intermunicipal hubs in rural areas with intermediate social vulnerability | 5 | 0.03 |
| 2a | Low populated, peripherical rural settlements with high household density and high social vulnerability | 2237 | 7.56 |
| 2b | Low populated peripherical rural settlements with socially vulnerable aged communities | 525 | 0.54 |
| 2c | Low populated peripheral rural settlements with moderately socially vulnerable aged communities | 543 | 1.16 |
| 3a | Medium-Low populated peri-urban settlements in rural areas with socially vulnerable aged communities | 457 | 1.1 |
| 3b | Medium-Low populated, high household density peri-urban settlements in rural areas with high social vulnerability | 220 | 1.47 |
| 3c | Medium-Low populated peri-urban settlements in rural areas with high social vulnerability | 955 | 4.13 |
| 4a | Medium-populated intermunicipal hubs with intermediate social vulnerability | 20 | 0.64 |
| 4b | Medium-populated intermunicipal hubs with high social vulnerability | 15 | 0.64 |
| 5 | Low Populated peri-urban suburbs with high social vulnerability | 615 | 2.92 |
| 6 | Medium-populated peri-urban and peripheric suburbs with moderate social vulnerability | 89 | 1.17 |
| 7 | Peripheric suburbs medium-low populated with high social vulnerability | 347 | 5.14 |
| 8a | Peri-urban cities with very high social vulnerability | 63 | 2.35 |
| 8b | Peri-urban cities with intermediate social vulnerability | 103 | 2.22 |
| 9a | Major urban hubs with low social vulnerability | 31 | 15.89 |
| 9b | Major urban hubs with intermediate social vulnerability | 20 | 6.61 |
| 10a | Medium-populated towns with high social vulnerability | 429 | 13.2 |
| 10b | Medium-populated towns with intermediate social vulnerability | 1286 | 33.23 |

*Archetype 1* represents rural settlements characterized by low population, low population density and medium-low social vulnerability, functioning as intermunicipal hubs. Notably, only few municipalities fall into this archetype, because it represents a small minority of urban hubs in rural areas with very low population (< 500 inhabitants). The majority of human settlements archetypes in peripheral and rural areas (i.e., those with low population density) are characterized by low population and medium to high social vulnerability (e.g., *archetypes 2a*, *2b*, *3a*, *3b* and *3c*). *Archetype 2a* includes peripherical, low populated rural areas (< 2000 inhabitants) with high social vulnerability, mainly due to a medium-high aging index, high low educational index, high unemployment rate and high crowding index. *Archetype 2b* represents sparsely populated rural, aging communities, exhibiting the highest average aging index and low educational index, along with significantly high proportion of buildings in poor conditions and low percentage of female employed, all contributing to a high SoVI value. In contrast, *archetype 2c*, which also represents low populated peripheral settlements characterized by very high aging index, shows lower SoVI value thanks to the lower percentage of commuters, a lower low educational index and higher proportion of female employed.

*Archetypes 3a*, *3b* and *3c* represent low populated, socially vulnerable settlements in peri-urban areas, exhibiting high low educational index, high percentage of unemployed (*3b*), high percentage of buildings in bad state of preservation (3a and 3b), high crowding index (*3b* and *3c*) and high proportion of foreign resident (3c). *Archetypes 4a* and *4b* are medium-populated intermunicipal hubs with slightly different level of social vulnerability: medium-low social vulnerability the former, medium-high social vulnerability the latter. While both show quite high crowding index, *archetype 4b* shows higher crowding index, high unemployment rate and medium-high percentage of buildings in poor conditions.

Low populated peri-urban suburbs are represented by *archetype 5* and show high social vulnerability, primarily due to medium-high unemployment rate, high percentage of commuters and high crowding index. *Archetype 6* corresponds to medium-populated peri-urban and peripheric suburbs with relatively favourable demographic and socio-economic conditions, benefiting from medium-high educational level (i.e., medium-low value of low educational index), low percentage of commuters, high percentage of female employed, that compensate for the medium-high value of buildings in bad state of preservation and high crowding index. *Archetype 7*, representing medium-low populated peripheral suburbs, shares the same average values of *archetype 6* regarding buildings in poor conditions and crowding index. However, it is characterized by higher percentage of commuters and higher percentage of unemployed, leading to higher SoVI.

*Archetypes 8a* represents densely populated peri-urban settlements with the highest social vulnerability, driven by the highest percentage of unemployed, the highest percentage of buildings in poor conditions and the highest crowding index. In addition, this archetype also shows high low educational index and low percentage of female employed. *Archetype 8b* also represents densely populated peri-urban settlements but with lower social vulnerability (higher percentages of female employed, lower percentage of buildings in bad state of preservation), despite being characterized by the highest percentage of commuters.

*Archetypes 9a* and *9b* correspond to the largest densely populated cities, yet with contrasting degree of social vulnerability: *archetype 9a* has low SoVI (the lowest), attributed to the lowest value of low educational index, the highest percentage of female employed, together with low percentage of buildings in poor conditions; *archetype 9b* has intermediate social vulnerability, due to the higher percentage of unemployed, higher percentage of buildings in poor conditions and higher crowding index. *Archetypes 10a* and *10b* represent medium-populated towns (with medium-population density) that can function as hubs or be located in peri-urban and peripheral areas. These towns exhibit relatively high social vulnerability, influenced by high percentage of unemployed (*10a*), medium-high percentage of commuters (*10b*), relatively low percentage of female employed (*10a*), high percentage of buildings in bad state of preservation (*10b*) and high crowding index.

From a geographical perspective, we found that many low populated peripheric and peri-urban settlements in rural areas (*archetypes 2b*, *2c*, *3a*, *3b* and *3c*) are primarily located in northern region of Piemonte (20%) and Lombardy (15%). *Archetypes 2a*, which also represents low populated peripherical rural areas, includes several municipalities in Sardinia (12%), Calabria (9%), Lombardy (9%) and Piemonte (7%) regions. *Archetype 4a* ("*Medium-populated intermunicipal hubs with intermediate social vulnerability*") is exclusively found in the northern regions (i.e., Piemonte, Lombardy, Veneto, Liguria, Tuscany and Marche). Conversely, *Archetype 4b* ("*Medium-populated intermunicipal hubs with high social vulnerability*") is present only in the southern regions, specifically Abruzzo, Campania, Apulia and Sicily. Many municipalities (48%) represented by *archetype 5* ("*Low Populated peri-urban suburbs with high social vulnerability*") are in Lombardy region, while a notable percentage of municipalities represented by *archetype 6* ("*Medium-populated peri-urban and peripheric suburbs with moderate social vulnerability*") are in Trentino (20%), Tuscany (11%) and Basilicata (11%) regions. *Archetype 7* ("*Peripheric suburbs medium-low populated with high social vulnerability*") includes a high concentration of municipalities in Lombardy (27%) and Sicily (15%) regions.

"*Peri-urban cities with very high social vulnerability*" (*archetype 8a*) can be found exclusively in the southern regions, with almost all cases located in Campania (61 out of 63 municipalities). In contrast, "*Peri-urban cities with intermediate social vulnerability*" (*archetype 8b*) are concentrated in the northern regions, specifically in Lombardy (99%) and Piemonte (1%). Similarly, "*Major urban hubs with low social vulnerability*" (*archetype 9a*) are in the northern part of the country, covering Piemonte, Valle d'Aosta, Lombardy, Trentino, Veneto, Friuli-Venezia Giulia, Liguria, Emilia-Romagna, Tuscany and Lazio regions, while "*Major urban hubs with intermediate social vulnerability*" (*archetype 9b*) predominantly located in the southern regions, including Molise, Campania, Apulia, Sicily and Sardinia regions. *Archetype 10a* ("*Medium-populated towns with high social vulnerability*") is also primarily found in the southern regions (93%), while *archetype 10b* ("*Medium-populated towns with intermediate social vulnerability*") is distributed across the entire country, with a significant percentage in Lombardy (27%) and Veneto (16%) regions.

## 6.1    Archetypes' vulnerability profiles

Composite indices are widely used to measure multidimensional concepts, as they enable the integration of various sub-indicators representing different dimensions that lack a common unit of measurement (Nardo et al., 2008). Social vulnerability and community resilience are often quantified through composite indices (e.g., Cutter et al., 2003; Frigerio et al., 2018; Bruneau et al., 2003; Marin Ferrer et al., 2017).

To investigate the level of exposure and vulnerability associated with each identified archetype, this study adopts a composite index-based framework. We define an Impact Susceptibility Index (ISI) which describes the potential for experiencing adverse consequences given existing vulnerabilities and exposure levels, without implying the occurrence of a specific hazard. The construction of the composite indicator involves four main stages: selection of sub-indicators, normalization, choice of aggregation method, and assignment of weights to the sub-indicators. The indicators used are those applied in the cluster analysis and described in Sections 2.1 to 2.5. Normalization—required to make the variables comparable and suitable for aggregation—is carried out by assigning categorical scores to each indicator, following approaches used in previous studies (e.g., Greiving et al., 2006). Scores range from 1 to 3, where 1 indicates low exposure or vulnerability and 3 indicates high exposure or vulnerability, hence contributing more to the susceptibility to impact for the given variable. For example, peripheral areas are considered the most vulnerable due to their greater distance from

essential services and are therefore assigned a score of 3. Peri-urban areas receive a score of 2, and urban hubs are assigned a score of 1. Similarly, since high population density is linked to greater physical vulnerability, cities are scored as 3, towns and suburbs as 2, and rural areas as 1. The highest population class (municipalities with over 250000 inhabitants) is also assigned the highest exposure and vulnerability score, while the lowest class (less than 5000 inhabitants) receives the lowest score. In terms of social vulnerability, three categories—high, medium, and low—are defined based on the Social Vulnerability Index range (0.84–2.01, see Table 4), and scores are assigned accordingly. The final ISI for each municipality is obtained by summing the individual scores for each vulnerability dimension (e.g., Greiving et al., 2006), and therefore range between 4 and 10. Figure 10 displays the resulting ISI at the municipal level and the average ISI for each archetype.

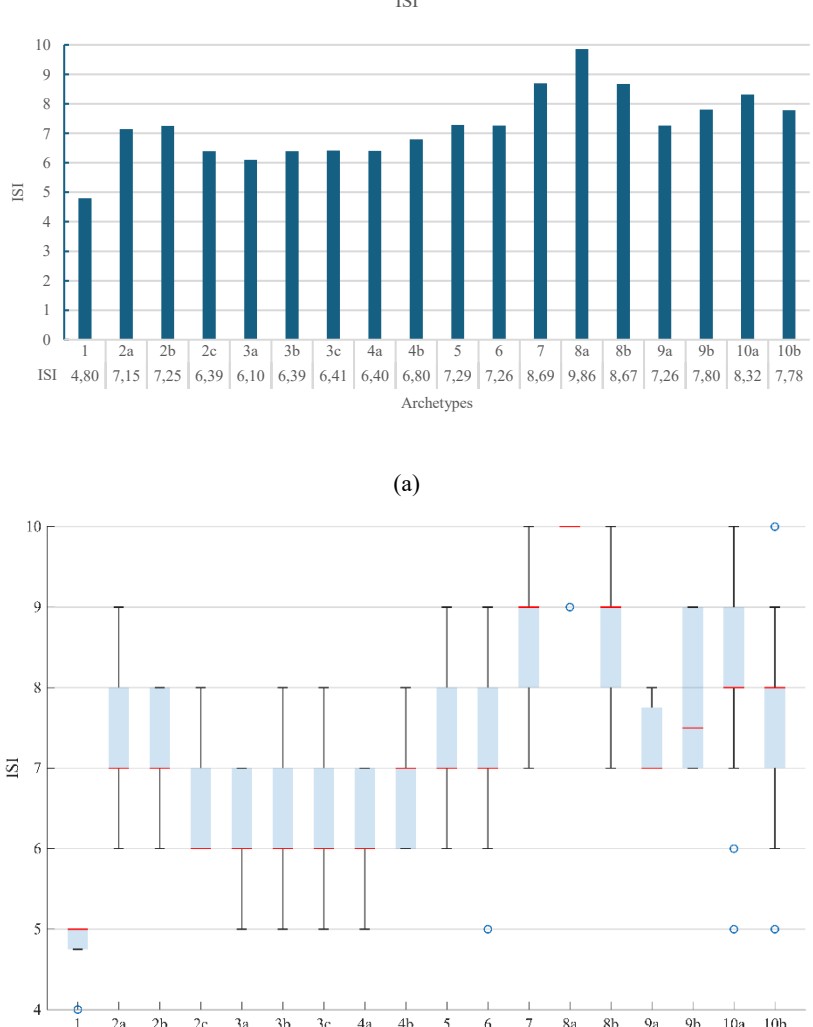

(a)

(b)

**Figure 10 – Average ISI value for each archetype (a); boxplot showing full distribution of ISI values, median (line in red), and outliers (blue circles) (b).**

The highest average ISI is observed for *Archetype 8a* (mean ISI = 9.86), which includes densely populated peri-urban municipalities characterized by very high social vulnerability. Other archetypes with notably high average ISI values include *Archetype 7* (mean ISI = 8.69), characterized by their relative remoteness (100% peripheral municipalities), medium-high population density (100% classified as towns and suburbs) and high social vulnerability; *Archetype 8b* (mean ISI = 8.67), marked by high population density (cities), peri-urban location and medium social vulnerability; and *Archetype 10a* (mean ISI = 8.32), largely driven by poor accessibility to services (only 12% of municipalities are classified as hubs), medium-high population and high social vulnerability. In several cases, social vulnerability is the primary driver of high ISI values, as observed in Archetypes 8a, 10a, 7 and Archetypes 2a, 2b and 2c. The latter (i.e., Archetypes 2a, 2b, and 2c) share the same geographic and demographic profiles yet differ in ISI values - with 2a and 2b showing higher ISI than 2c - due solely to differences in their SoVI scores. In this context, SoVI emerges as the only influencing factor driving

ISI variation among these archetypes. Conversely, for other archetypes, demographic and geographic characteristics play a more significant role in shaping ISI outcomes. For instance, Archetype 9b presents the lowest SoVI but a relatively high ISI, which can be attributed to its high population density. In contrast, Archetype 3b shows one of the highest mean SoVI scores, second only to Archetype 9b, but results in a relatively low ISI, primarily due to its low population density and geographic remoteness.

The box plot in Figure 10b illustrates the distribution of ISI values across the identified urban archetypes, providing insights into both central tendency and internal variability; the red and black lines represent the median and min-max values of ISI for each archetype, blue boxes the dispersion (±standard deviation) and the dots are outliers, in case they are present. Archetype 1 exhibits the lowest median ISI and minimal variability, suggesting a consistently low level of exposure and vulnerability across its municipalities. In contrast, Archetype 8a displays the highest median ISI with a very narrow spread, indicating strong internal homogeneity and high susceptibility to impacts. The few outliers with an ISI value of 9 highlight minor deviations but do not significantly affect the overall pattern. Archetype 9b shows the greatest dispersion, reflecting a high degree of internal heterogeneity. This wide variability suggests the presence of municipalities with both relatively low and high ISI values within the same archetype, potentially complicating uniform policy interventions. Archetypes 3a, 3b, and 3c present identical median, mean, and interquartile ranges, which align with their shared geographic and demographic features, as well as similar SoVI scores (see Table 3). However, their comparable SoVI estimates result from distinct socio-economic compositions, as discussed in Section 5, underscoring the multidimensional nature of social vulnerability.

Figure 11 highlights that many municipalities with high ISI values are concentrated in the regions of Apulia, Sicily and Lombardy, with average regional ISI values of 7.8, 7.7, 7.6, respectively. In details, 37% of Apulia's municipalities fall under *Archetype 10a*; 25% of Sicily's municipalities are categorized as *Archetypes 10a* while 14% belong to *Archetype 7*. In Lombardy, 22% of municipalities belong to *Archetype 10b*, 19% to *Archetype 5* and 14% to *Archetype 2a*. These archetypes all show medium-high average ISI values: 7.78, 7.29, and 7.15, respectively. Overall, the ISI tends to be higher in southern regions of Italy, with an average value of 7.3, compared to 6.9 for central and northern regions. The lowest average VI values are observed in the Valle d'Aosta (ISI = 6.4) and Piedmont (ISI = 6.7) regions.

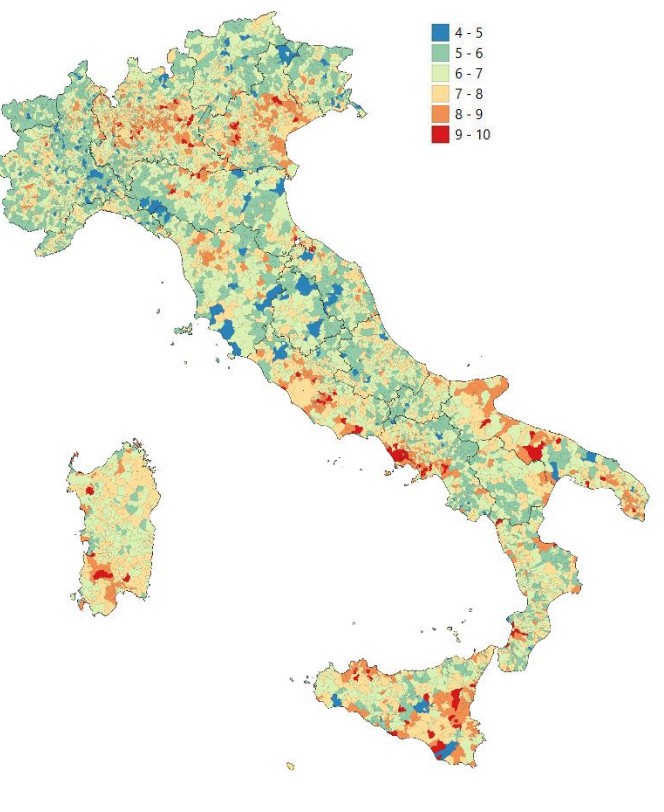

**Figure 11 - Map of ISI value at municipal level.**

## 7 Discussion

The proposed study of human settlements archetypes leverages the framework and guidelines set forth by Piemontese et al. (2022) to ensure a robust and reliable archetype analysis, focusing on six dimensions of validity: conceptual validity, construct validity, internal validity, empirical validity, external validity, and application validity. The proposed archetypes conform to each of these dimensions as follows.

Conceptual Validity is achieved by ensuring the research problem and questions are scientifically sound and relevant to real-world issues. In this study we addressed the need to categorize urban and rural areas based on geographic, demographic, and socio-economic factors to understand urban/rural vulnerabilities better. By focusing on these pertinent aspects, this study aligns with the conceptual framework and reflects real-world challenges faced by urban and rural settlements in Italy.

Construct Validity involves the careful selection of attributes that define the archetypes, ensuring their connection to the conceptual framework. We meticulously selected attributes relevant to vulnerability of urban/rural systems and their potential exposure to different hazards. These attributes are justified based on existing literature, ensuring indicators are theoretically and empirically linked to several vulnerability dimensions, thereby reinforcing the construct validity (Diogo et al., 2023; Nagel et al., 2024).

Internal Validity is maintained through the rigorous application of hierarchical and partitioning clustering methods. In previous studies, such as Bilalova et al. (2025), internal validity has been addressed through a transparent and replicable methodology, incorporating widely used validity metrics which measures how similar an object is to its own cluster compared to other clusters like silhouette scores, and evaluating both within-cluster and between-cluster cohesion. Similarly, Nagel et al. (2024) assessed internal validity by testing cluster robustness using *R* packages *NbClust* and *clValid*. The *NbClust* package supports the determination of the optimal number of clusters by computing and comparing multiple internal validity indices (e.g., silhouette score, Dunn index), while the *clValid* package enables the evaluation of clustering stability and comparative performance across different algorithms (e.g., k-means, hierarchical clustering). Building on these approaches, internal validity in this study is ensured through: (i) the determination of the optimal number of clusters using established internal validity indices—specifically, the inconsistency coefficient and the WCD; (ii) the assessment of cluster robustness, by repeating the partitioning clustering procedure multiple times with randomized initial centroids and selecting the best-performing result based on WCD, thus reducing sensitivity to initialization; (iii) the comparison of clustering algorithms, by applying both hierarchical and partitioning methods and evaluating their performance using WCD and ICD to identify the most internally coherent solution. This combination of techniques ensures methodological rigor, reproducibility, and robustness in the clustering process, thereby addressing the internal validity dimension as recommended in the literature (Piemontese et al., 2022).

Empirical validity in archetype analysis is commonly supported through various means, including stakeholder surveys (Nagel et al., 2024), the integration of diverse data sources at different spatial resolutions, and cross-comparisons of archetyping approaches at multiple scales (Diogo et al., 2023). It may also be demonstrated through consistency with prior empirical observations or theoretical expectations (Bilalova et al., 2025). However, validating archetypes' vulnerability profiles against observed impacts or risk outcomes remains challenging. For example, many historical impact datasets—such as those from EM-DAT—are available only at national or regional levels and include only events meeting specific severity criteria. As a result, they often exclude smaller-scale, yet locally significant, events, introducing both a selection bias and a scale mismatch that limit their utility for validating local-level archetypes. Furthermore, expected impact outputs from risk assessments are typically model-driven, emphasizing hazard intensity and physical exposure, while often overlooking the broader dimensions of vulnerability (Cardona et al., 2012). These limitations highlight the need for improved access to fine-grained, georeferenced impact data and the potential value of complementing quantitative validation with qualitative or stakeholder-informed insights at the local level. Empirical validity in our research is partially supported by stakeholder engagement-based risk storylines, as outlined e.g. in Marciano et al. (2024). Marciano et al. (2024) present an exploratory case study using a participatory approach to develop multi-risk storylines, illustrating the cascading effects of a heatwave followed by intense rainfall in two Italian urban contexts: a peri-urban area and a metropolitan area. Findings reveal that peri-urban settlements face limited emergency resources and higher infrastructure failure risks, while metropolitan hubs have stronger emergency systems but face coordination challenges in managing large-scale events. The study highlights the varying levels of vulnerability across different archetypes. While these elements contribute to the empirical grounding of the archetypes, we acknowledge that empirical validation remains a limitation of this study. Further studies should explore the impacts of natural disasters on different archetypes, revealing key differences in vulnerability and response capabilities across the considered urban contexts.

External Validity assesses the generalizability of archetypes beyond the studied cases. It is typically addressed by applying archetypes across multiple regions and evaluating the consistency of resulting patterns across different scales (e.g., Diogo et al., 2023; Nagel et al., 2024) or linking archetypes to theoretical expectations or global typologies (e.g., Bilalova et al., 2025). While this study acknowledges the challenge of fully satisfying this dimension, given that the identified archetypes are specific to the Italian context and broader applicability requires further investigation, it also provides a foundation for generalization. The identification of archetypes across diverse Italian regions, combined with the careful selection of relevant variables and the use of a replicable methodology, may serve as a valuable reference for archetype-based analyses in other national or regional settings, particularly within Europe. Notably, many of the variables adopted in this study, such as degree o urbanisation, population class, and census-based demographic and socio-economic indicators, are also available at comparable spatial resolutions through Eurostat, EUROPOP, or pan-European datasets such as Urban Atlas, CORINE Land Cover, and GHSL (Global Human Settlement Layer). Similarly, the degree of urban centeredness, while constructed using national criteria in Italy, relates closely to the concept of accessibility and service availability, which can be captured using EU-wide datasets on transport networks, healthcare access, and educational infrastructure. Therefore, the consistent use of open-source and harmonized data sources enhances the potential for applying the methodology beyond the Italian context, fostering comparative analyses and supporting the construction of cross-country urban and rural archetypes within Europe.

Application Validity evaluates the practical usefulness of the archetypes. This dimension can be addressed emphasizing practical applications of archetypes in policy, planning, and governance, for instance, by presenting results to government officials and researchers, guiding inform local policy discussions, with archetypes guiding differentiated policy interventions (Nagel et al., 2024). Section 6.1 illustrates the potential of urban/rural archetypes to enhance risk communication through the assignment of a simplified impact susceptibility index to each identified archetype. Additionally, the exploratory case study presented in Marciano et al. (2024) highlights how these archetypes can support stakeholder engagement by informing the development of multi-risk storylines. By categorizing human settlements into distinct archetypes, it becomes possible to assess how different hazard scenarios may unfold in each context, considering their specific vulnerabilities, exposure levels, and adaptive capacities. This structured approach enables policymakers to design tailored interventions and resilience strategies based on specific vulnerability profiles. However, to further strengthen resilience planning and develop targeted mitigation measures, it is crucial to consider not only exposure and vulnerability but also hazard data for each archetype—particularly the level of exposure of a settlement to various natural hazards. Although in this study we did not yet integrate hazard information, there is a clear need for future research to incorporate this aspect and conduct GIS-based analyses for a more comprehensive assessment of risk (e.g., Tocchi et al., 2024).

## 8    Conclusion

This study presents a set of archetypes for urban and rural settlements in Italy, based on geographic, demographic and socio-economic factors that cover different vulnerability dimensions. Using a two-step cluster analysis, ten broad archetypes were first defined according to structural features (e.g., location, size, density), further refined into 18 nested archetypes to account for socio-economic diversity.

The proposed archetypes were developed by applying the six dimensions of validity outlined by Piemontese et al. (2022), offering a robust and replicable methodology for vulnerability-oriented archetype analysis. While several of these validity dimensions was successfully addressed (conceptual, construction, internal and application validity), empirical and external validity were only partially addressed. Conceptual, construct, and internal validity are robustly established through scientifically sound research questions, careful attribute selection, and rigorous clustering methods. Empirical validity of proposed archetypes may be hardly satisfied, as discussed, due to the lack of fully integrated social and institutional vulnerability data, as well as limitations of risk modelling. External validity remains an open challenge: while the archetypes are context-specific to Italy, the use of open and harmonized data sources (e.g., Urban Atlas, CORINE Land Cover, Eurostat demographic indicators, GHSL datasets) enhances the potential for replicating the methodology in other European contexts, fostering future comparative studies. Application validity was demonstrated by linking each archetype to an Impact Susceptibility Index, providing a tool for prioritizing areas for risk reduction strategies. The archetypes also offer structured support for developing multi-risk storylines and informing resilience planning efforts.

Despite some limitations, this study provides a valuable framework for simplifying complex urban and rural vulnerability patterns. It lays a strong foundation for both scientific advancements and practical applications in the field of multi-risk assessment, resilience planning, and targeted policy design. Defining urban and rural archetypes based on vulnerability factors may help identify areas with higher susceptibility to natural hazards and socio-economic challenges, supporting

better resource allocation for disaster preparedness and response. It also highlights critical areas for future research. In particular, integrating hazard-specific exposure data and further empirical validation through observed impact data are needed to fully realize the potential of archetype-based approaches in disaster risk management and climate change adaptation.

**Data Availability**

The complete dataset describing the urban archetypes associate to each municipality is available in:
Tocchi, G., Pittore, M., & Polese, M. (2025). Italian urban archetypes (Tocchi et al., 2025) (1.0) [Data set]. Zenodo. https://doi.org/10.5281/zenodo.14888733.

**Acknowledgements**

This study was carried out within the RETURN Extended Partnership and received funding from the European Union Next-GenerationEU (National Recovery and Resilience Plan – NRRP, Mission 4, Component 2, Investment 1.3 – D.D. 1243 2/8/2022, PE0000005)

**Author contribution**

All authors contributed to the conceptualization and design of the methodology. G. T. carried out the formal analysis, led the research and investigation process, managed data visualization and presentation, and was responsible for writing the original draft along with subsequent editing. M. P. and M. P. also contributed to the writing, review, and editing of the manuscript.

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
