# Peer review of "Identifying urban and rural settlement archetypes: clustering for enhanced risk-oriented exposure and vulnerability analysis"

_EGUsphere, 2025_

## Referee Comment (RC2)

**Review of Manuscript egusphere-2025-908**

This manuscript uses clustering algorithms to define settlement archetypes for the Italian territory based on variables known to be relevant metrics for risk/resilience from previous studies. The outcome is a set of 10 first-level archetypes, some of which are sub-divided and lead to a total of 18 categories, as well as a classification of Italian municipalities into each of these categories. As the authors well state, the definition of such archetypes has the potential to be relevant for supporting decision-making processes associated with vulnerability and risk reduction.

However, the manuscript appears as incomplete work in this regard. Clustering algorithms are numerical methods that group individuals (in this case, municipalities) as per their similarity with respect to the variables considered. The algorithms know nothing about the meaning of these variables. The manuscript is lacking: (1) a more in-depth interpretation of what it means or what consequences may stem from the fact that municipalities with some differences in certain attributes end up grouped together, as well as (2) a demonstration of some sort that these 18 archetypes are indeed associated with relevant risk metrics. In other words, do these groups actually make sense when confronted against risk analyses (single or multi-hazard)? Is the "vulnerability profile" (as the authors phrase it) meaningful? The fact that the variables selected as input for the clustering are known to be relevant from previous studies does not guarantee that the grouping is relevant and consequential in the risk domain, or suited for designing risk reduction actions.

I believe this manuscript should address this major shortcoming as well as the following main comments before it is considered for publication.

**Main Comments**

1. Title: The archetypes defined are not just urban. Please erase "urban" from the title and throughout the body of the manuscript.

2. As stated above, the major shortcoming of this manuscript is the lack of in-depth interpretation of the results and confrontation against risk data or models that demonstrate its relevance and meaning for the purpose that the authors state this work has. Here are some thoughts that aim at helping with developing the work further:

    a) The clustering was carried out using a series of parameters known to be relevant to vulnerability/resilience. The outcome is groups of data points that are "similar" to each other with respect to those same parameters. Without a comparison against actual risk metrics, it is not proven that these archetypes mean anything to risk. Moreover, there is no conclusion with respect to how exactly decision makers should use them. Which archetypes should be prioritised for in-depth risk assessments, for example?

    b) What would happen if you just enumerated all possible combinations of your categorical parameters (which could also be used by decision makers to prioritise areas of intervention)? What would the map look like in that case? Does the difference between this "full enumeration" map and the one you obtained say anything about the relevance of the clustering?

    c) Some combinations of parameters simply do not exist in Italy. This is one of the reasons that may prevent the exact same archetypes from being applicable to other regions (which is a potential limitation raised in lines 732-736 of the manuscript). A comparison across countries is only relevant in terms of risk metrics, otherwise the comparison is just about knowing how many combinations of parameters exist in each country.

    d) A clustering algorithm is agnostic to the meaning of the input data. The analysis could have included variables that are not relevant to risk at all, and they would have still come up in the clustering. This is not to say that the risk metrics should be included in the clustering. There is value in carrying out the clustering using the variables you used (because these are variables easily obtained from open datasets, because they can be used as proxies, etc), but the output needs to be interpreted in terms of risk metrics in order to be meaningful for vulnerability/risk reduction.

3. The selection of variables to include is difficult to follow. During my first read I could not understand why some variables from Table 1 were missing in Table 2. It only became clear as I kept on reading (around line 180, i.e. ~50 lines after Table 2 is mentioned in the text). Moreover, line 189 mentions 19 attributes, but Table 2 contains 15. It is only clear in Table 3 that some of the socio-economic parameters of Table 2 are further subdivided. Moreover, in Table 2 it is not clear how some of the parameters are measured (e.g., "quality of the buildings"). Then there is the issue that the number of numerical variables is further reduced due to their correlation (section 4.1), which is correct. But then the final variables used are only listed within the text (lines 386-388), which gets lost with respect to what the reader can easily identify from the tables. In lines 366-367 it is stated that the population variable is used as classes, which was already implicit in section 3.4 (consider moving the spirit of what is said in lines 361-367 to section 3.4). All these issues result in the manuscript feeling messy. Please consider unifying some of the three tables (Tables 2 and 3 are good candidates), using more columns or bullet points to sub-classify items, perhaps marking the 8 numerical variables used in the end, and indicating which variables are categorical and which numerical. Please make it clear from the beginning that Table 1 is just a discussion of potential parameters to include, but not the parameters used.

4. There is some imbalance in the level of detail provided for the first level of clustering (section 4.4), which is associated with the categorical variables, and that of the second level (section 4.5), which refers to the numerical variables. Please consider expanding section 4.5, perhaps including informative figures/plots.

5. Section 5 (results) is merely descriptive. Deeper insights on the meaning of the resulting clusters and their spatial distribution are completely missing. Lines 642-683 mostly repeat what is already said in Tables 4 and 5. Lines 684-705 mostly describe the spatial distribution of the archetypes with respect to Italian regions but (i) the regions are not marked in any map (please do not assume all readers will know where these regions are) and, most importantly, (ii) no insights are provided with respect to what this spatial distribution may mean or imply.

6. Section 6 (discussion) only refers to how the methodology used complies with the guidelines of Piemontese et al. (2022), in some cases inaccurately or obscurely. In some detail:

   a) L720-722: Employing WCD and ICD does not ensure the reliability or robustness of the clustering process. Again, too much faith is being put on the numerical algorithms. How is the accuracy (with which the dataset is represented, as per the wording in L721-722) measured?

   b) L730-731: While the authors imply that confronting the archetypes against risk data/models is outside of the scope of the paper, the classification work per se does not hold relevant meaning without this confrontation.

   c) L732-736: As stated above, comparing the clustering output of different countries without going into the risk domain is just a comparison of how these sets of variables are combined and spatially distributed in different countries. There would be no further implication.

   d) L737-738: The sentence "This study *demonstrates* the potential of urban archetypes in developing risk storylines, enhancing risk communication, and supporting stakeholder engagement" is simply not true. Where exactly in the manuscript are these three points demonstrated?

7. In the last paragraph of the conclusions (L759-761): "*its findings offer a valuable tool for policy makers to design targeted interventions based on specific vulnerability profiles*": The paper does not demonstrate that the archetypes are indeed associated with specific vulnerability profiles, it just assumes they are. Again, the paper needs to somehow show that the archetypes are indeed associated with vulnerability profiles that make sense for policy making.

**Minor Comments/Edits**

1. L13: "complexity in an otherwise too broad problem".

2. L24: "Over the last few decades".

3. L43: Exposure is not "intended". Do you mean understood? Defined?

4. L43-45: Please replace the semicolons ( ; ) by commas.

5. L58: It says "we assume", but perhaps you mean "we hypothesize"?

6. L61: Erase "We note that". It is not necessary.

7. L83-85: Are capitals really needed for "design", "analysis" and "application"?

8. L95: No comma after "defined".

9. Table 1: It says "GPD per capita" instead of "GDP per capita".

10. L161: "complicates evacuation efforts and strains emergency response resources".

11. L167: Consider breaking the paragraph here and starting a new one to talk about the socio-economic factors.

12. L169: "highlight that the elderly".

13. L192: "proposed a grid-based approach".

14. L201: "adopting  the abovementioned" (remove "to").

15. Fig. 1: Please consider changing the colour of peri-urban areas. The map on the right shows six labels, but only three classes of urban centeredness degree are considered in the analysis. "Hub" and "municipal hub" can be seen as different shades of red of the final class "urban hubs", while the intermediate, peripheral and ultra-peripheral inland areas are indifferent shades of blue, so that is clear. However, peri-urban areas are in orange, which makes them very similar to the "urban hubs" class. A different colour for peri-urban areas would make it easier to visualise the final three classes used. Parentheses in the text (lines 243-245) could clarify this. E.g. "namely urban hubs (represented…, shades of red in Fig. 1), peri-urban areas (new colour) and inland areas (shades of blue)".

16. L258: "lead to the model becoming".

17. L259: "correlation that may exist".

18. Table 2 vs line 377 and Fig. 3: Is it population aged under 15 or 14 (different numbers in different places).

19. Table 2 vs line 311: Should it be "total population aged over 30" under "high educational index"?

20. L344: Clustering methods *per se* do not "ensure" that observations within the same group are highly similar. It is their goal, but it is not guaranteed. They are just numerical methods. They will return some sort of clustering no matter what you provide as input. Please re-phrase.

21. L354-355: "More detailed information".

22. L359: "is crucial for enhancing the quality of the clustering".

23. L376-383: The strong negative correlation between the proportion of employed and un

24. Fig. 3, labels (list 1-14):

    - (1) "Proportion of population under 14" (or another alternative, the current grammar is not right).

    - (2) "Proportion of population over 65" (or another alternative, the current grammar is not right).

    - (12) Replace "components" with "members".

    - (14) "Residents" (plural), not "resident" (singular).

    In general, please aim for consistency between the labels used in this figure and in table 3.

25. Fig. 3: It would be easier to interpret the matrix if the colour scale was based on three colours instead of two (right now, from yellow to blue, passing through green). This would make values around zero more visible.

26. Fig. 3: Is it correct that the correlation between features 7 and 8 is so low, numerically? These features are the proportion of employed and unemployed among the working-age population. Are they not one minus the other (1 – employed = unemployed) and, if not (because of some other category), are they not at least highly negatively correlated?

27. L406: Should the subscript of the x variables be "l" instead of "k", to be consistent with Eq. (1) and the text?

28. L515: "The attributes' importance for clustering is evaluated adopting the simplified procedure".

29. L516: "to assess their contribution".

30. L519: "WCD as the performance metric".

31. L526: Do you mean cluster 2, instead of 6?

32. L564: "The lower is the WCDl" (erase "is").

33. Table 4 is colour-coding the SoVI. Please mention the colours in the text (lines 614-616).

34. Table 4:

- "Building poor state"? You used "poor" before. Please be consistent.

- "Foreigners" or "Foreign residents".

- Clusters 2a and 4b: "Settlements with high social vulnerability".

35. Table 4: As they are phrased, sub-clusters 2a (settlements with high social vulnerability) and 2b (settlements with aged population with high social vulnerability) overlap, with 2b seemingly a sub-category of 2a. If 2a explicitly excludes aged populations, then the label should reflect this

36. L696-697: "in the southern regions, with almost all cases located".

---

## Author Comment (AC1)

Reviewer 1

**Topic and key findings of manuscript**

Thank you for the opportunity to review this manuscript, which analyzes 18 municipal archetypes for the Italian territory, incorporating geographic, demographic, and socio-economic characteristics. Deriving archetypes for risk, exposure and vulnerability analysis is a highly relevant topic for scientific investigation. The paper is therefore an interesting contribution to the interdisciplinary debate on how to construct such archetypes. However, the arguments and the way they are presented need to be revised before publication.

**(a) Title:** consider revising the title. First of all, you speak of "urban" archetypes but your analysis includes several rural archetypes as well. Maybe "municipal" would be more fitting. I suggest you revise this throughout the manuscript. Secondly, you speak of multi-risk, but do not really elaborate on that. I would suggest you rather revise the title so that it is clear that your main goal is to construct the archetypes.

**Response**: Thanks for this comment. The term "*urban*" has been replaced with "*urban and rural*", as also rural archetypes are included in this study. It has been reviewed throughout the manuscript. Moreover, the term "multi-risk" has been removed, and the title has been changed from: "*Identifying urban settlement archetypes: clustering for enhanced multi-risk exposure and vulnerability analysis*" to "*Identifying urban and rural settlement archetypes: clustering for enhanced risk-oriented exposure and vulnerability analysis*". In addition, to underline the fact that both urban and rural archetypes are addressed in this study, the following text highlighted in bold has been added in the introduction:

*"The level to which urban settings are prone to the negative impacts of one or multiple hazards is also known as urban vulnerability (Thywissen, 2006), and its assessment is particularly challenging, as cities are intricate systems composed of interdependent networks of built environments, infrastructure, and social systems (Koren et al., 2017). The concentration of assets and people may increase potential losses, while dynamic interactions between individual components that enable efficient system performance can lead to cascading failures. In addition, urban areas are often exposed to multiple hazards, such as earthquakes, floods, heatwaves, each interacting with the built environment and human activities in different ways. **Rural settlements, on the other hand, may experience different forms of vulnerability, often related to geographic isolation, limited access to emergency services and infrastructure, lower institutional capacity, and demographic challenges such as aging population, which can significantly hinder preparedness and recovery**. This complex interplay explains also why often non-extreme hazards can lead to severe consequences, while extreme events in other contexts may not result in disasters (Lavell et al., 2012)"*

**(b) Abstract:** The abstract should be revised to incorporate the changes detailed below.

**Response**: The abstract has been modified according to changes made. The new version of the abstract is reported below:

*"Identification of risks and vulnerabilities in **urban and rural areas** is crucial for supporting local authorities in disaster risk reduction and climate change adaptation. Moreover, comparison of risk assessments across **different areas** may help effective allocation of adaptation funding towards more resilient and sustainable **communities**. The distinct physical, social, economic, and environmental characteristics of a **settlement**, along with the relevance of impending hazards, determine the level of risk and vulnerability faced by its residents. While the results of risk assessments will vary from one **settlement** to another, using general **settlement typologies** (e.g. coastal cities, dryland cities, and inland or high-altitude cities) can effectively support the understanding of risk in relation to its key drivers, helping to segmentate the complexity in otherwise too broad problem (Dickson et al., 2012).*

*This study aims to reduce complexity in risk assessment of **urban/rural settlements** at regional and national scale, ensure a baseline for comparison and identify potential hotspots in **risk assessment frameworks**. We*

*propose a clustering methodology that groups **human settlements** based on open-source data, used as proxies of urban vulnerability and exposure. Applying two widely used clustering techniques, we define 18 **urban and rural archetypes** for the Italian territory, incorporating geographic, demographic, and socio-economic characteristics. These archetypes satisfy multiple validity dimensions of archetype analysis (Piemontese et al., 2022) and can serve as a valuable tool for policymakers. By providing a structured understanding of **human settlements** vulnerability profiles, they support the design of targeted interventions and resilience strategies tailored to specific risk conditions."*

**(b) Structure of the manuscript:** I recommend revising the structure to improve conciseness (see detailed comments below). The introduction, discussion, and conclusion are relatively brief, whereas the materials and methods section is quite extensive and could be streamlined. Additionally, clearer section headings, especially in sections 2 and 3, would help distinguish between the introduction and the materials section.

**Response**: Thanks for this comment. The introduction has been revised, including also concepts presented in section 2 – which has been removed to have a better distinction between the different part of the paper, i.e., introduction, material and methods. See also response to following comment for details about introduction's revisions. In addition, section 3 title has been changed from "*Key indicators of urban vulnerability*" to "*Selection of key indicators of vulnerability dimensions*".

**(c) Introduction:** The introduction would benefit from further elaboration. Specifically, I suggest clarifying how archetypes enhance the understanding of exposure and vulnerability in this context. Additionally, since archetype analysis can take various forms, it is important to highlight how previous studies have approached archetypes and to clearly define your own understanding of the concept. Providing a brief explanation of how you apply the concept and implement it with your data—before presenting the archetypes in Chapter 5—would improve clarity. A figure of the framework could help with clarity. Further, I suggest to include the research question more prominently in this section.

**Response**: The introduction has been revised to address the reviewer's comment by: (i) clearly stating the research question early on; (ii) elaborating on how archetypes enhance the understanding of exposure and vulnerability; (iii) clarifying different interpretations of archetypes and explicitly stating the one used in this study; (iv) briefly explaining how the concept is operationalized in this work. To further improve the clarity and readability of the paper, the section where each step of the proposed procedure is described has been referenced. The revised version of the introduction is provided below, with the added or modified sentences highlighted in bold:

[revised manuscript text omitted]

**(d) Materials and Methods:** I appreciate the thorough justification and explanation of the datasets. However, this section could be more concise. For instance, Tables 1 and 2 might be combined. Additionally, it would be helpful to clarify why you differentiate between main and sub-clusters, as this reflects your understanding of archetypes. The methods section is quite extensive and could be streamlined. I also recommend making it clearer why you use different clustering approaches and what distinguishes their outcomes. Since testing these differences appears to be a key finding, it should be addressed not only here but also in the introduction, results, and discussion sections.

**Response**: Thanks for this comment. The following key points raised have been addressed: (i) reorganization of paper's tables; (ii) a clearer justification for the distinction between main and sub-clusters; (iii) an explanation regarding the use of different clustering approaches.

(i) Table 2 and Table 3 have been removed. The indicators selected for urban settlement clustering have been described throughout the text and reported in a new summary Table - provided in Section 4.1. (now, Section 3.1; see response to comment e). Specifically, the attributes adopted in the study have been directly linked to the indicators and the vulnerability dimensions presented in Table:

*"Table 1 presents a list of key indicators commonly used in literature to assess each dimension of vulnerability mentioned.*

*In our work we focused on a selection of indicators, expectedly linked with different vulnerability dimensions, and namely: altimetric zone, centeredness degree, urban degree, residential population and social vulnerability indicators. The altimetric zone of a settlement, which refers to their elevation and topographical features, can be considered a proxy of access to the main services – or equally distance to service centres* **(institutional vulnerability, see Table 1)**. *Accessibility of services of general interest can be particularly challenging in certain contexts (e.g. mountain regions, islands) due to their geomorphological and settlement structure conditions (Bertram et al., 2023). These accessibility issues can also complicate evacuation efforts and the delivery of emergency services during a disaster. Likewise, urban centeredness degree, which reflects the spatial characteristics and distribution of urban areas, is associated with the availability of public services and the level of spatial connectedness, as it measures the distance and travel time to major service centres* **(institutional vulnerability, see Table 1)**. *The degree of urban centeredness significantly influences the response and resilience of urban systems by affecting resource availability, infrastructure robustness, community networks, and emergency preparedness (Giuliano & Narayan, 2003; Schwanen et al., 2004). Ensuring effective access to essential public services, such as healthcare and education, is challenging even under normal circumstances. However, it becomes even more crucial during crises like natural disasters, when the demands on these services and their operating conditions become significantly more complex (Fan et al., 2022; Loreti et al., 2022; Tariverdi et al., 2023). The level of peripherality of the areas with respect to the network of urban centres influence may determine not only difficulties of access to basic services but also lower quality of life of citizens and their level of social inclusion (Oppido et al., 2023).*

*Residential population and urban degree are linked to exposure and physical vulnerability dimensions, and specifically to population density* **(physical vulnerability, see Table 1)**. *Residential population significantly influences the exposure to natural hazards, determined not only by the higher presence of people and housing, but also of infrastructure, production capacities, species or ecosystems, and other tangible human assets in places and settings that could be adversely affected by one or multiple hazards. Higher population not only increases the potential for human and property losses, but also complicate evacuation efforts, and strain emergency response resources (Zhao et al., 2017). The degree of urbanization is often used to classify areas into cities, urban areas, and rural areas based on criteria such as population density, concentration of human activities, and built environment (Balk et al., 2018; United Nations, 2018). Indeed, highly urbanized densely populated areas are more likely to experience greater damage, congestion, and strain on resources during emergencies. It affects the capacity for evacuation and accessibility to essential services, due to dense infrastructure, complex urban layouts and the potential for cascading failures in infrastructure (Kendra et al., 2008; Lall & Deichmann, 2012).*

*Finally, social vulnerability indicators include those parameters that influence both* **social and economic vulnerability**. *Past events highlight that elderly may be more vulnerable due to reduced mobility, poor health, and communication challenges (ARDALAN et al., 2010; Carnelli & Frigerio, 2017; Cutter et al., 2003), while education levels can heighten vulnerability to natural hazards influencing risk perception and awareness, knowledge, and skills related to disaster preparedness (Alexander, 2012; Wachinger et al., 2013). Still, minority groups, including migrants, and ethnic communities, often face heightened social vulnerability, especially in high-risk areas, due to language barriers and communication challenges that can hinder access to critical emergency information (Carnelli & Frigerio, 2017; Walter Gillis et al., 2012).* **The complete list of socio-economic indicators considered includes age, dependency ratio, level of education, family structures,**

*commuting rate, quality of buildings, race/ethnicity, employment rate, percentage of women in the alterworkforce (Table 2).*

*It is worth mentioning that we only consider indicators for which publicly available and authoritative data exist at the municipal level. For example, since GDP per capita is only available at national, regional, or provincial scales, it is not included in this study. Similarly, many building characteristics affecting physical vulnerability are either difficult to detect or unavailable at the municipal scale (e.g., structural system and earthquake-resistant design level; Tocchi et al., 2022). Moreover, building vulnerability indicators often vary depending on the type of hazard (Kappes et al., 2012), making it challenging to collect all relevant information for multiple hazards across Italy. For these reasons, only population density and general building quality are considered in this study. Indicators suggested for the environmental vulnerability dimension are not included due to data limitations as well. For instance, municipal-level air pollution data in Italy is limited, as such data is only available for major cities with monitoring stations.*

*Data on urban degree, urban centeredness degree, altimetric zone, social vulnerability factors used herein are primarily sourced from ISTAT (Italian National Institute of Statistics). All data are collected at the municipal level, aligning with the administrative boundaries adopted for the analysis. The dataset includes 7960 objects, representing the 7960 Italian municipalities, and 19 attributes (both numerical and categorical) related to the vulnerability factors outlined in sections 3.1 through 3.5."*

The Table listing all variable considered for the clustering is reported at the end of Section 4.1. "Data pre-processing" (now, Section 3.1; see response to comment *e*). It is reported below:

**Table 2 – Variable used in cluster analysis.**

| Variable | Type | Vulnerability dimension |
|---|---|---|
| Urban degree | Categorical | Physical |
| Urban centeredness degree | Categorical | Institutional |
| Population class | Categorical | Physical |
| Altimetric zone | Categorical | Institutional |
| Aging index | Numerical | Social |
| Low educational index | Numerical | Social |
| Unemployed | Numerical | Economic |
| Commuting rate | Numerical | Social |
| Female employed | Numerical | Economic |
| Quality of buildings | Numerical | Social |
| Crowding index | Numerical | Social |
| Foreign resident | Numerical | Social |

(ii)  The use of a two-step clustering approach—differentiating between main and sub-clusters—is motivated by the need to balance simplicity with depth. The initial "broad" classification reduces complexity and provides a baseline for comparison, offering a clear and interpretable framework to capture key structural differences among urban and rural settlements (e.g., population size, density, location). This step helps organize the diversity of settlements into coherent categories based on fundamental geographic and demographic characteristics. In the second step, refining these broad archetypes using socio-economic variables allows for a deeper analysis of vulnerability within structurally similar contexts. This nested approach enhances the understanding of intra-group variability and supports more targeted risk assessments and tailored policy interventions. This rationale has also been added to the revised version of the introduction for greater clarity (see response to comment *c*).

(iii)  The use of different clustering techniques was already justified in the introduction: "*Using two clustering techniques allows for cross-validation of results and helps capture different patterns in the data, enhancing the robustness and reliability of the identified archetypes*" and in the discussion section: "*Internal Validity is maintained through the rigorous application of hierarchical and partitioning clustering methods. We employed WCD and ICD to select the most suitable clustering approach, ensuring the reliability and robustness of the clustering process*" while specific sub-section is dedicated to the comparison

of the outcomes of the two clustering algorithms, i.e., section 4.4.3 (now section 4.2., see response to comment *e*). An additional description of benefit deriving from the adoption of two different clustering techniques has been added in the methodological section (section 3 "*Cluster analysis*"):

*"The adoption of two different clustering techniques serves to enhance the robustness, reliability, and interpretability of the archetypes identified in this study. Each method has distinct strengths and analytical advantages, which, when combined, allow for a more comprehensive exploration of patterns in the data. For instance, hierarchical clustering is particularly useful for exploring data structures without the need to predefine the number of clusters. It produces a dendrogram that visually represents nested groupings and their relationships, offering insights into how clusters evolve as dissimilarity thresholds change. This is especially valuable for understanding the hierarchical nature of urban/rural systems and guiding the selection of an appropriate number of clusters. On the other hand, partitioning clustering requires the number of clusters to be predefined, but it typically performs better with larger datasets, producing compact, well-separated clusters when appropriately parameterized. It is computationally efficient and more suitable for refining clusters, especially when working with both categorical and numerical data types. Using both techniques enables cross-validation of clustering outputs, ensuring consistency and increasing confidence in the identified archetypes. Discrepancies between methods can highlight ambiguous or transitional settlement types, while convergences confirm stable, well-defined clusters."*

**(e) Results:** The distinction between the materials, methods, and results sections could be clearer. Since you refer to Italian regions, I suggest adding their borders to the results map to aid readability. Additionally, consider adjusting the colors in Figure 9, as some archetypes are difficult to differentiate.

**Response**: Thanks for this comment. The following key points raised have been addressed: (i) the reorganization of the materials, methods, and results sections; (ii) the modifications of colors in Figure 9.

(i) The Methods and Results sections have been clearly distinguished in the revised manuscript. Specifically, the former subsection titled "*Clustering based on demographic and geographic features*" has been updated as section and renamed to "*First-level analysis: clustering based on physical and institutional vulnerability parameters*", while the subsection "*Nested clustering*" has been updated as section named "*Second-level analysis: nested clustering based on socio-economic parameters*". As a result, the revised paper now includes three distinct sections:

- Section 3: "*Cluster analysis*" – dedicated to the description of the methodology
- Section 4: "*First-level analysis: clustering based on physical and institutional vulnerability parameters*" – presenting the results of the first step of the archetype analysis
- Section 5: "*Second-level analysis: nested clustering based on socio-economic parameters*" – presenting the results of the second step.

The final definition and discussion of the urban and rural settlement archetypes for Italy are provided in Section 6: "*Urban and rural settlement archetypes in Italy*".

(ii) Colors of figure 9 have been slightly modified, to better differentiate the broad archetypes. The old and the updated figures are reported below. Moreover, on the right part of Figure 9, a map showing the border and names of the Italian regions is reported

[Figure]

**(f) Discussion:** The discussion could be expanded, as it is currently quite brief. It should engage more with existing literature and clarify how your choice of data and methods influenced the results. Since you refer to dimensions of validity, I recommend elaborating on this aspect by discussing how these dimensions are addressed in relation to the literature. Currently, the claim that your archetypes meet validity requirements lacks sufficient support.

**Response:** The discussion section has been expanded. Specifically, discussion about how validity dimensions are addressed in literature has been added. The new version of this section is reported below (with modified or added text highlighted in bold):

*"The proposed study of human settlements archetypes leverages the framework and guidelines set forth by Piemontese et al. (2022) to ensure a robust and reliable archetype analysis, focusing on six dimensions of validity: conceptual validity, construct validity, internal validity, empirical validity, external validity, and application validity. The proposed archetypes conform to each of these dimensions as follows.*

[revised manuscript text omitted]

*Application Validity evaluates the practical usefulness of the archetypes.* **This dimension can be addressed emphasizing practical applications of archetypes in policy, planning, and governance, for instance, by presenting results to government officials and researchers, guiding inform local policy discussions, with archetypes guiding differentiated policy interventions (Nagel et al., 2024). Section 6.1 illustrates the potential of urban/rural archetypes to enhance risk communication through the assignment of a simplified impact susceptibility index to each identified archetype. Additionally, the exploratory case study presented in Marciano et al. (2024) highlights how these archetypes can support stakeholder engagement by informing the development of multi-risk storylines.** *By categorizing human settlements into distinct archetypes, it becomes possible to assess how different hazard scenarios may unfold in each context, considering their specific vulnerabilities, exposure levels, and adaptive capacities. This structured approach enables policymakers to design tailored interventions and resilience strategies based on specific vulnerability profiles. However, to further strengthen resilience planning and develop targeted mitigation measures, it is crucial to consider not only exposure and vulnerability but also hazard data for each archetype—particularly the level of exposure of a settlement to various natural hazards. Although in this study we did not yet integrate hazard information, there is a clear need for future research to incorporate this aspect and conduct GIS-based analyses for a more comprehensive assessment of risk (e.g., Tocchi et al., 2024)."*

Moreover, application validity has been further improved and presented in a new section of the paper (6.1), reported below.

*"6.1 Archetypes' vulnerability profiles*

[revised manuscript text omitted]

**g) Conclusion:** Ensure that the conclusion aligns with the preceding sections. Either here or in the discussion, clarify what is needed to refine the archetypes and how they enhance the understanding of exposure and vulnerability.

**Response**: Thanks for this comment. The revised version of the conclusion, which incorporates the suggested modifications and clearly outlines the strengths and limitations of the proposed study, is provided below:

*"This study presents a set of archetypes for urban and rural settlements in Italy, based on geographic, demographic and socio-economic factors that cover different vulnerability dimensions. Using a two-step cluster analysis, ten broad archetypes were first defined according to structural features (e.g., location, size, density), further refined into 18 nested archetypes to account for socio-economic diversity.*

*The proposed archetypes were developed by applying the six dimensions of validity outlined by Piemontese et al. (2022), offering a robust and replicable methodology for vulnerability-oriented archetype analysis. While several of these validity dimensions was successfully addressed (conceptual, construction, internal and application validity), empirical and external validity was only partially addressed. Conceptual, construct, and internal validity are robustly established through scientifically sound research questions, careful attribute selection, and rigorous clustering methods. Empirical validity of proposed archetypes may be hardly satisfied, as discussed, due to the lack of fully integrated social and institutional vulnerability data. External validity remains an open challenge: while the archetypes are context-specific to Italy, the use of open and harmonized data sources (e.g., Urban Atlas, CORINE Land Cover, Eurostat demographic indicators, GHSL datasets) enhances the potential for replicating the methodology in other European contexts, fostering future comparative studies. Application validity was demonstrated by linking each archetype to an Impact Susceptibility Index, providing a tool for prioritizing areas for risk reduction strategies. The archetypes also offer structured support for developing multi-risk storylines and informing resilience planning efforts.*

*Despite some limitations, this study provides a valuable framework for simplifying complex urban and rural vulnerability patterns. It lays a strong foundation for both scientific advancements and practical applications in the*

*field of multi-risk assessment, resilience planning, and targeted policy design. Defining urban and rural archetypes based on vulnerability factors may help identify areas with higher susceptibility to natural hazards and socio-economic challenges, supporting better resource allocation for disaster preparedness and response. It also highlights critical areas for future research. In particular, integrating hazard-specific exposure data and further empirical validation through observed impact data are needed to fully realize the potential of archetype-based approaches in disaster risk management and climate change adaptation."*

**Minor issues:**

**Figures 4 and 6:** Both figures currently have the same caption. To avoid confusion, clarify that they represent different methods and specify the distinctions in their captions.

**Response:** Thanks for this feedback. To avoid confusion, the caption of the figures has been modified as follows:

*(i)* **Figure 1 - Representativeness of clusters resulting from hierarchical clustering in terms of urban degree (a), urban centeredness degree (b), population class (c) and altimetric zone (d).**

*(ii)* **Figure 6 - Representativeness of clusters resulting from partitioning clustering in terms of urban degree (a), urban centeredness degree (b), population class (c) and altimetric zone (d).**

**Introduction and Abstract:** You mention single and multiple hazards but do not elaborate on them in the main sections. Since your focus is primarily on exposure and vulnerability, consider toning down these references for consistency.

**Response:** Introduction and abstract have been modified accordingly. See also response to comments (b) and (c).

**Tables and Formatting:** The placement of tables is inconsistent, with some splitting across pages in a way that affects readability. The editorial team should ensure that, where possible, tables fit within a single page to improve clarity.

**Response:** Position of tables (specifically, table 1) has been modified in order to avoid splitting of the table across pages.

**Line 526:** Is the reference to cluster 6 correct here? The number does not align with Figure 5. Please verify and ensure consistency between the text and the figure.

**Response:** Thanks for this comment. It was a typo. In Line 526 the correct reference is to cluster 2. It has been corrected in the new version of the paper.

**Line 287:** write remaining instead of remain

**Response:** The term "remain" has been replaced by "remaining".

**Figure 3:** The colors used in this figure may be difficult to interpret for readers with color blindness. Consider using a blue-white-red color scheme to improve accessibility.

**Response:** Colors of the figure have been modified. Please find below Figure 3 revised with new color scheme.

[Figure]

**Correlation Matrix**

1. Proportion of under 14 aged
2. Proportion of over 65 aged
3. Aging index
4. Dependency ratio
5. High educational index
6. Low educational index
7. Proportion of employed
8. Proportion of unemployed
9. Proportion of commuters
10. Proportion of female employed
11. Proportion of buildings in poor condition
12. Proportion of families with 5 or more components
13. Crowding index
14. Proportion of foreign resident

**Figures:** Please ensure consistency in your referencing throughout instead of alternating between, for example, "Fig" and "Figure."

**Response:** The term "Fig." has been replaced by "Figure" throughout the paper.

*References*

[revised manuscript text omitted]

---

## Author Comment (AC2)

This manuscript uses clustering algorithms to define settlement archetypes for the Italian territory based on variables known to be relevant metrics for risk/resilience from previous studies. The outcome is a set of 10 first-level archetypes, some of which are sub-divided and lead to a total of 18 categories, as well as a classification of Italian municipalities into each of these categories. As the authors well state, the definition of such archetypes has the potential to be relevant for supporting decision-making processes associated with vulnerability and risk reduction.

However, the manuscript appears as incomplete work in this regard. Clustering algorithms are numerical methods that group individuals (in this case, municipalities) as per their similarity with respect to the variables considered. The algorithms know nothing about the meaning of these variables. The manuscript is lacking: (1) a more in-depth interpretation of what it means or what consequences may stem from the fact that municipalities with some differences in certain attributes end up grouped together, as well as (2) a demonstration of some sort that these 18 archetypes are indeed associated with relevant risk metrics. In other words, do these groups actually make sense when confronted against risk analyses (single or multi-hazard)? Is the "vulnerability profile" (as the authors phrase it) meaningful? The fact that the variables selected as input for the clustering are known to be relevant from previous studies does not guarantee that the grouping is relevant and consequential in the risk domain, or suited for designing risk reduction actions.

I believe this manuscript should address this major shortcoming as well as the following main comments before it is considered for publication.

**Main comments**

1. Title: The archetypes defined are not just urban. Please erase "urban" from the title and throughout the body of the manuscript.

**Response:** Thanks for this comment. The term "*urban*" has been replaced with "*urban and rural*", as also rural archetypes are included in this study. It has been reviewed throughout the manuscript.

2. As stated above, the major shortcoming of this manuscript is the lack of in-depth interpretation of the results and confrontation against risk data or models that demonstrate its relevance and meaning for the purpose that the authors state this work has. Here are some thoughts that aim at helping with developing the work further:

   a. The clustering was carried out using a series of parameters known to be relevant to vulnerability/resilience. The outcome is groups of data points that are "similar" to each other with respect to those same parameters. Without a comparison against actual risk metrics, it is not proven that these archetypes mean anything to risk. Moreover, there is no conclusion with respect to how exactly decision makers should use them. Which archetypes should be prioritised for in-depth risk assessments, for example?

   **Response:** Thanks for this important comment. To ensure consistency, the empirical validation of archetypes ideally requires comparison with actual risk metrics that incorporate all relevant dimensions of vulnerability used in the archetype analysis—such as social and institutional vulnerability. However, such comparisons are often hindered by practical limitations. For instance, observed disaster impact data at fine spatial resolutions (e.g., municipal level) are frequently incomplete, inconsistent, or unavailable, especially in contexts where disaster reporting is not standardized or centralized. Additionally, many historical datasets, like EM-DAT, typically provide information only at national or regional scales, limiting their utility for local-level validation. Furthermore, existing risk assessments

often rely on models focused primarily on physical exposure and hazard intensity, while overlooking critical vulnerability dimensions such as social or institutional factors. These aspects—like population aging, poverty, and limited access to services—can significantly influence disaster outcomes by delaying emergency response, worsening health impacts, and hampering recovery efforts (Cardona et al., 2012; Cutter et al., 2003; Marin Ferrer, 2017). Some studies adopted composite risk indices that integrate multiple dimensions of risk and vulnerability to validate urban archetypes (e.g., Riach et al., 2023). Despite several examples of composite risk index are available in literature, many of these are primarily designed for national or regional-level assessments and are less suitable for high-resolution, multi-hazard risk evaluation (e.g., Greiving et al., 2006; Marin Ferrer, 2017). Other approaches, such as the U.S. national risk index by Zuzak et al.(2022), despite offering integrated multi-hazard assessments using high-resolution data, could be hardly applied in countries like Italy due to the unavailability of similarly detailed and harmonized data, especially for resilience metrics. Also, the study proposed by Tocchi et al. (2025) - which introduces a multi-hazard risk index specifically developed for the Italian context, integrating multiple hazards with physical and social vulnerability dimensions, as well as exposure – is not suitable for assessing empirical validity, since it partly overlaps in the choice of exposure and vulnerability indicators and also includes hazard descriptors.

Given the challenges associated with validating this dimension, we acknowledge that empirical validation remains a limitation of the present study. Further research is necessary to better understand and demonstrate the differences in vulnerability and response capacities across the various urban contexts analyzed. However, this work offers a valuable foundation for both future scientific investigations and practical applications. By systematically identifying and classifying urban settlements into archetypes based on a range of (accurately selected) vulnerability-related factors - using a replicable, data-driven approach grounded in open-source information - this study contributes to simplifying the complexity of urban risk landscapes. It provides a structured framework that can support more targeted and effective risk reduction strategies, inform urban resilience planning, and facilitate stakeholder engagement (as demonstrated also by the study conducted by Marciano et al., 2024). To underline this limitation, the following text has been added in the discussion section:

*"Empirical validity in archetype analysis is commonly supported through various means, including stakeholder surveys (Nagel et al., 2024), the integration of diverse data sources at different spatial resolutions, and cross-comparisons of archetyping approaches at multiple scales (Diogo et al., 2023). It may also be demonstrated through consistency with prior empirical observations or theoretical expectations (Bilalova et al., 2025). However, validation of archetypes' vulnerability profiles with observed impacts or risk analysis outcomes is often hard since (i) many historical impact datasets (such as those from EM-DAT) are only available at national or regional scales, limiting their usefulness for local-level archetype validation; (ii) expected impact results from risk assessments often rely on models that focus on physical exposure and hazard intensity, while frequently neglecting other vulnerability dimensions (Cardona et al., 2012). For instance, social and institutional vulnerabilities—such as aging populations, poverty, and limited access to services—are often overlooked in standard risk models, despite their significant role in influencing disaster outcomes. These factors can delay emergency responses, worsen health impacts, and hinder recovery, ultimately amplifying both direct and indirect losses (Cutter et al., 2003; Cardona et al., 2012; Marin-Ferrer et al., 2017). Empirical validity in our research is partially supported by stakeholder engagement-based risk storylines, as outlined in Marciano et al. (2024). Marciano et al. (2024) present an exploratory case study using a participatory approach to develop multi-risk storylines, illustrating the cascading effects of a heatwave followed by intense rainfall in two Italian urban contexts: a peri-*

*urban area and a metropolitan area. Findings reveal that peri-urban settlements face limited emergency resources and higher infrastructure failure risks, while metropolitan hubs have stronger emergency systems but face coordination challenges in managing large-scale events. The study highlights the varying levels of urban vulnerability across different archetypes. While these elements contribute to the empirical grounding of the archetypes, we acknowledge that empirical validation remains a limitation of this study. Further studies should explore the impacts of natural disasters on different archetypes, revealing key differences in vulnerability and response capabilities across the considered urban contexts."*

Moreover, to underline the usefulness of this study in decision-making process, a simplified index (named Impact Susceptibility Index) is assigned to each archetype. It is presented in a new section of the paper (6.1), reported below:

[revised manuscript text omitted]

b.  What would happen if you just enumerated all possible combinations of your categorical parameters (which could also be used by decision makers to prioritise areas of intervention)? What would the map look like in that case? Does the difference between this "full enumeration" map and the one you obtained say anything about the relevance of the clustering?

**Response**: In theory, enumerating all possible combinations of the four categorical parameters used as input in our clustering analysis—urban centeredness degree (3 classes), urban degree (3 classes), population (3 classes), and altimetric zone (3 classes)—would yield a total of 3 × 3 × 3 × 3 = 81 possible combinations. This "full enumeration" approach would result in a highly granular classification, where each municipality is assigned a unique label corresponding to its specific combination of characteristics. Furthermore, these combinations would have a very heterogeneous distribution in size, given to implicit constraints across the parameters (e.g. few highly populated municipalities are located in high altimetric zones, as also highlighted by the comment "C").

While such a map might be useful for purely descriptive or diagnostic purposes, especially for decision-makers aiming to filter or prioritize interventions based on specific parameter configurations, it would lack the ability to generalize across similar contexts. In contrast, the clustering approach aggregates municipalities into a reduced number of archetypes that share common multi-dimensional characteristics, even when those characteristics are not identical across all variables. This enables the identification of broader, more interpretable settlement typologies that reflect meaningful patterns and shared vulnerability profiles, which is particularly valuable in risk-informed decision-making.

Moreover, the difference between the "full enumeration" map and the clustering-based archetype map highlights the relevance of clustering in simplifying complexity and

enhancing interpretability. Such a comparison can be inferred from Figures 4 and 5 in the paper, which illustrate how the clusters represent the underlying attributes. These figures highlight correlations between variables that would not be discernible through a simple full enumeration map. While the full enumeration map would display a highly fragmented spatial pattern with many singletons or very small groups, the clustering outcome promotes synthesis by grouping municipalities with similar profiles, facilitating communication and operationalization of results. Therefore, the clustering does not simply reproduce the combinations of categorical inputs but instead detects underlying structures and similarities that may not be immediately apparent in the raw categorical space. This supports the added value of our approach for both scientific analysis and practical applications.

c.  Some combinations of parameters simply do not exist in Italy. This is one of the reasons that may prevent the exact same archetypes from being applicable to other regions (which is a potential limitation raised in lines 732-736 of the manuscript). A comparison across countries is only relevant in terms of risk metrics, otherwise the comparison is just about knowing how many combinations of parameters exist in each country.

**Response**: Thanks for this comment.   The authors acknowledge the limitations in generalizing the archetypes beyond the studied context. Indeed, the identified archetypes are specific to the Italian case, and their broader applicability would require further investigation. However, the identification of archetypes across diverse Italian regions— along with the careful selection of relevant variables and the use of a replicable methodology—may serve as a valuable reference for archetype-based analyses in other national or regional settings, particularly within Europe. The limitations and potential of applying the proposed archetype analysis to other regions are discussed in the Discussion section:

*"External Validity assesses the generalizability of archetypes beyond the studied cases. It is typically addressed by applying archetypes across multiple regions and evaluating the consistency of resulting patterns across different scales (e.g., Diogo et al., 2023; Nagel et al., 2024) or linking archetypes to theoretical expectations or global typologies (e.g., Bilalova et al., 2025). While this study acknowledges the challenge of fully satisfying this dimension, given that the identified archetypes are specific to the Italian context and broader applicability requires further investigation, it also provides a foundation for generalization. The identification of archetypes across diverse Italian regions, combined with the careful selection of relevant variables and the use of a replicable methodology, may serve as a valuable reference for archetype-based analyses in other national or regional settings, particularly within Europe. Notably, many of the variables adopted in this study, such as urban degree, population class, and census-based demographic and socio-economic indicators, are also available at comparable spatial resolutions through Eurostat, EUROPOP, or pan-European datasets such as Urban Atlas, CORINE Land Cover, and GHSL (Global Human Settlement Layer). Similarly, the urban centeredness degree, while constructed using national criteria in Italy, relates closely to the concept of accessibility and service availability, which can be captured using EU-wide datasets on transport networks, healthcare access, and educational infrastructure. Therefore, the consistent use of open-source and harmonized data sources enhances the potential for applying the methodology beyond the Italian context, fostering comparative analyses and supporting the construction of cross-country urban and rural archetypes within Europe."*

d.  A clustering algorithm is agnostic to the meaning of the input data. The analysis could have included variables that are not relevant to risk at all, and they would have still come up in the clustering. This is not to say that the risk metrics should be included in the clustering. There is value in carrying out the clustering using the variables you used (because these are

variables easily obtained from open datasets, because they can be used as proxies, etc), but the output needs to be interpreted in terms of risk metrics in order to be meaningful for vulnerability/risk reduction.

**Response**: All variables included in the analysis have been deemed relevant to vulnerability/resilience and exposure, since have been identified through a detailed literature review. This accurate selection of variables ensures indicators are theoretically and empirically linked to several vulnerability dimensions, thereby reinforcing the construct validity, as also shown in other archetypes studies (e.g., Diogo et al., 2023; Nagel et al., 2024). This has been also underlined in the discussion session:

*Construct Validity involves the careful selection of attributes that define the archetypes, ensuring their connection to the conceptual framework. We meticulously selected attributes relevant to vulnerability of urban systems and their potential exposure to different hazards. These attributes are justified based on existing literature, ensuring indicators are theoretically and empirically linked to several vulnerability dimensions, thereby reinforcing the construct validity (Diogo et al., 2023; Nagel et al., 2024)."*

Moreover, the outputs have been also interpreted in terms of susceptibility to impacts using the Impact Susceptibility Index, and the limitations related to the comparison with real-world impact data or the outcomes of existing risk analyses have been clearly highlighted in the discussion section. Please refer to the response to comment 2a for further details.

3. The selection of variables to include is difficult to follow. During my first read I could not understand why some variables from Table 1 were missing in Table 2. It only became clear as I kept on reading (around line 180, i.e. ~50 lines after Table 2 is mentioned in the text). Moreover, line 189 mentions 19 attributes, but Table 2 contains 15. It is only clear in Table 3 that some of the socio-economic parameters of Table 2 are further subdivided. Moreover, in Table 2 it is not clear how some of the parameters are measured (e.g., "quality of the buildings"). Then there is the issue that the number of numerical variables is further reduced due to their correlation (section 4.1), which is correct. But then the final variables used are only listed within the text (lines 386-388), which gets lost with respect to what the reader can easily identify from the tables. In lines 366-367 it is stated that the population variable is used as classes, which was already implicit in section 3.4 (consider moving the spirit of what is said in lines 361-367 to section 3.4). All these issues result in the manuscript feeling messy. Please consider unifying some of the three tables (Tables 2 and 3 are good candidates), using more columns or bullet points to sub-classify items, perhaps marking the 8 numerical variables used in the end, and indicating which variables are categorical and which numerical. Please make it clear from the beginning that Table 1 is just a discussion of potential parameters to include, but not the parameters used.

**Response:** Thanks for this comment. To improve clarity, Table 2 and Table 3 have been removed. It has been specified that Table 1 only reports a list of indicators commonly used in literature to assess the different vulnerability dimensions, while all the variables selected for the clustering are mentioned in the text in the section dedicated to data section ("*Selection of key indicators of vulnerability dimensions*") and connected to vulnerability indicators presented in Table 1. More specifically, the text has been modified as follows:

*"Table 1 presents a list of key indicators commonly used in literature to assess each dimension of vulnerability mentioned.*

*In our work we focused on a selection of indicators, expectedly linked with different vulnerability dimensions, and namely: altimetric zone, centeredness degree, urban degree, residential population and social vulnerability indicators. The altimetric zone of urban settlements, which refers to their elevation and*

*topographical features, can be considered a proxy of access to the main services – or equally distance to service centres (institutional vulnerability, see Table 1). Accessibility of services of general interest can be particularly challenging in certain contexts (e.g. mountain regions, islands) due to their geomorphological and settlement structure conditions (Bertram et al., 2023). These accessibility issues can also complicate evacuation efforts and the delivery of emergency services during a disaster. Likewise, urban centeredness degree, which reflects the spatial characteristics and distribution of urban areas, is associated with the availability of public services and the level of spatial connectedness, as it measures the distance and travel time to major service centres (institutional vulnerability, see Table 1). The degree of urban centeredness significantly influences the response and resilience of urban systems by affecting resource availability, infrastructure robustness, community networks, and emergency preparedness (Giuliano & Narayan, 2003; Schwanen et al., 2004). Ensuring effective access to essential public services, such as healthcare and education, is challenging even under normal circumstances. However, it becomes even more crucial during crises like natural disasters, when the demands on these services and their operating conditions become significantly more complex (Fan et al., 2022; Loreti et al., 2022; Tariverdi et al., 2023). The level of peripherality of the areas with respect to the network of urban centres influence may determine not only difficulties of access to basic services but also lower quality of life of citizens and their level of social inclusion (Oppido et al., 2023).*

*Residential population and urban degree are linked to exposure and physical vulnerability dimensions, and specifically to population density (physical vulnerability, see Table 1). Residential population significantly influences the exposure to natural hazards, determined not only by the higher presence of people and housing, but also of infrastructure, production capacities, species or ecosystems, and other tangible human assets in places and settings that could be adversely affected by one or multiple hazards. Higher population not only increases the potential for human and property losses, but also complicate evacuation efforts, and strain emergency response resources (Zhao et al., 2017). The degree of urbanization is often used to classify areas into cities, urban areas, and rural areas based on criteria such as population density, concentration of human activities, and built environment (Balk et al., 2018; United Nations, 2018). Indeed, highly urbanized densely populated areas are more likely to experience greater damage, congestion, and strain on resources during emergencies. It affects the capacity for evacuation and accessibility to essential services, due to dense infrastructure, complex urban layouts and the potential for cascading failures in infrastructure (Kendra et al., 2008; Lall & Deichmann, 2012).*

*Finally, social vulnerability indicators include those parameters that influence both social and economic vulnerability. Past events highlight that elderly may be more vulnerable due to reduced mobility, poor health, and communication challenges (Ardalan et al., 2010; Carnelli & Frigerio, 2017;Cutter et al., 2003), while education levels can heighten vulnerability to natural hazards influencing risk perception and awareness, knowledge, and skills related to disaster preparedness (Alexander, 2012; Wachinger et al., 2013). Still, minority groups, including migrants, and ethnic communities, often face heightened social vulnerability, especially in high-risk areas, due to language barriers and communication challenges that can hinder access to critical emergency information (Carnelli & Frigerio, 2017; Walter Gillis et al., 2012). The complete list of socio-economic indicators considered includes age, dependency ratio, level of education, family structures, commuting rate, quality of buildings, race/ethnicity, employment rate, percentage of women in the workforce (Table 2). "*

Since the number of numerical variables is further reduced due to their correlation, to make clearer which variables are finally adopted in the clustering (and their type) the following table (the new Table 2) has been added in the paper. It is reported at the end of Section 4.1. "Data pre-processing" (now, Section 3.1; see response to comment *4*).

**Table 2 – Variable used in cluster analysis.**

| Variable | Type | Vulnerability dimension |
|---|---|---|
| Urban degree | Categorical | Physical |
| Urban centeredness degree | Categorical | Institutional |
| Population class | Categorical | Physical |
| Altimetric zone | Categorical | Institutional |
| Aging index | Numerical | Social |

| Low educational index | Numerical | Social |
|---|---|---|
| Unemployed | Numerical | Economic |
| Commuting rate | Numerical | Social |
| Female employed | Numerical | Economic |
| Quality of buildings | Numerical | Social/Physical |
| Crowding index | Numerical | Social |
| Foreign resident | Numerical | Social |

4. There is some imbalance in the level of detail provided for the first level of clustering (section 4.4), which is associated with the categorical variables, and that of the second level (section 4.5), which refers to the numerical variables. Please consider expanding section 4.5, perhaps including informative figures/plots.

**Response:** Section 4.4 included both a detailed description of the adopted methodology (i.e., the clustering techniques applied) and the corresponding results. This led to a more extensive discussion, whereas Section 4.5, which employed the same methodology already presented in Section 4.4, focused solely on presenting the results and was therefore shorter.

In the revised manuscript, we have clearly distinguished between the Methods and Results sections to improve structure and balance. Specifically:

- The subsection previously titled "*Clustering based on demographic and geographic features*" has been renamed "*First-level analysis: clustering based on physical and institutional vulnerability parameters*."
- The subsection "*Nested clustering*" has been updated to "*Second-level analysis: nested clustering based on socio-economic parameters*."

As a result, the revised paper now includes three distinct sections:

- Section 3: "*Cluster analysis*" – dedicated to the description of the methodology
- Section 4: "*First-level analysis: clustering based on physical and institutional vulnerability parameters*." – presenting the results of the first step of the archetype analysis
- Section 5: "*Second-level analysis: nested clustering based on socio-economic parameters*" – presenting the results of the second step.

The authors believe that this restructuring improves the clarity and coherence of the manuscript.

5. Section 5 (results) is merely descriptive. Deeper insights on the meaning of the resulting clusters and their spatial distribution are completely missing. Lines 642-683 mostly repeat what is already said in Tables 4 and 5. Lines 684-705 mostly describe the spatial distribution of the archetypes with respect to Italian regions but (i) the regions are not marked in any map (please do not assume all readers will know where these regions are) and, most importantly, (ii) no insights are provided with respect to what this spatial distribution may mean or imply.

**Response:** Thanks for this feedback. Section 5 (now Section 6; see response to comment 4) presents the results of the cluster analysis, and the characterization of the archetypes derived from both categorical and numerical variables used in the study. This section is essential for understanding the rationale behind the archetype names and how they relate to the attributes shown in Tables 4 and 5. Moreover, section 5 (now section 6) includes also a discussion on spatial distribution to reflect on regional differences in terms of vulnerability drivers and settlement patterns, offering meaningful insights into what the spatial distribution of archetypes implies in terms of regional risk governance and planning needs.

To better address your comment and enhance the interpretability of the results:

- A new overview map of Italy has been added next to Figure 9, showing the names and locations of all Italian regions, to assist readers who may be unfamiliar with the country's geography (see below).
- A new section has also been added to discuss the vulnerability profiles of the identified archetypes, providing a deeper interpretation of their characteristics and the implications for disaster risk and resilience (see also response to comment 2a).

[Figure]

Figure 9 – Broad urban archetypes across Italian territory. On the right the map with the Italian regions is reported.

6. Section 6 (discussion) only refers to how the methodology used complies with the guidelines of Piemontese et al.(2022), in some cases inaccurately or obscurely. In some detail:

   a. L720-722: Employing WCD and ICD does not ensure the reliability or robustness of the clustering process. Again, too much faith is being put on the numerical algorithms. How is the accuracy (with which the dataset is represented, as per the wording in L721-722) measured?

   **Response:** Thank you for the comment. We acknowledge that relying solely on numerical metrics such as Within-Cluster Distance (WCD) and Inter-Cluster Distance (ICD) may not fully ensure the reliability or robustness of the clustering process. However, these metrics are widely recognized in literature as important tools for assessing internal validity, by quantifying how well data points are grouped within clusters and separated from other clusters (Bilalova et al., 2025; Nagel et al., 2024). In addition to these metrics, we also evaluated the interpretability and coherence of the resulting clusters in relation to known patterns in demographic and geographic data. This qualitative cross-check helps reinforce the internal consistency of the archetypes.

   Regarding the term '*accuracy*' in L721–722, we refer to the degree to which the clusters reflect meaningful patterns in the dataset, based on both numerical cohesion/separation

and the interpretability of the resulting groups. To clarify this, we have revised the manuscript text accordingly:

*"Internal Validity is maintained through the rigorous application of hierarchical and partitioning clustering methods. In previous studies, such as Bilalova et al. (2025), internal validity has been addressed through a transparent and replicable methodology, incorporating widely used validity metrics which measures how similar an object is to its own cluster compared to other clusters like silhouette scores, and evaluating both within-cluster and between-cluster cohesion. Similarly, Nagel et al. (2024) assessed internal validity by testing cluster robustness using R packages NbClust and clValid. The NbClust package supports the determination of the optimal number of clusters by computing and comparing multiple internal validity indices (e.g., silhouette score, Dunn index), while the clValid package enables the evaluation of clustering stability and comparative performance across different algorithms (e.g., k-means, hierarchical clustering). Building on these approaches, internal validity in this study is ensured through: (i) the determination of the optimal number of clusters using established internal validity indices—specifically, the inconsistency coefficient and the WCD; (ii) the assessment of cluster robustness, by repeating the partitioning clustering procedure multiple times with randomized initial centroids and selecting the best-performing result based on WCD, thus reducing sensitivity to initialization; (iii) the comparison of clustering algorithms, by applying both hierarchical and partitioning methods and evaluating their performance using WCD and ICD to identify the most internally coherent solution. This combination of techniques ensures methodological rigor, reproducibility, and robustness in the clustering process, thereby addressing the internal validity dimension as recommended in the literature (Piemontese et al., 2022)."*

b. L730-731: While the authors imply that confronting the archetypes against risk data/models is outside of the scope of the paper, the classification work per se does not hold relevant meaning without this confrontation.

   **Response:** The authors have strengthened the practical relevance of this study for decision-making processes by calculating a simplified index—the Impact Susceptibility Index—for each identified archetype. This index measures the propensity for impacts based on the exposure and vulnerability characteristics of each archetype (see also response to comment 2a). The authors believe that this addition significantly enhances the satisfaction of the application validity dimension and offers a useful ranking to help prioritize archetypes in risk reduction strategies. At the same time, the authors acknowledge that fully satisfying the empirical validity dimension remains a limitation of this study. Nevertheless, the work provides a solid foundation for future scientific research and practical applications, successfully addressing the other validity dimensions (see also responses to comments 6a, 2a, and 2c). This issue has already been discussed in response to comment 2a.

c. L732-736: As stated above, comparing the clustering output of different countries without going into the risk domain is just a comparison of how these sets of variables are combined and spatially distributed in different countries. There would be no further implication.

   **Response:** The authors agree that comparing clustering outputs across different countries without connecting them to the risk domain may appear to be a purely descriptive exercise with limited implications. However, the authors believe such a comparison can still provide meaningful insights, particularly when the clustering is based on vulnerability-related indicators that are theoretically and empirically associated with disaster risk. By identifying whether similar vulnerability profiles (e.g., combinations of socio-economic, demographic, and geographic characteristics) emerge across diverse national contexts, we can begin to assess the universality or context-specificity of certain vulnerability patterns.

While our primary focus in this study is on the Italian context, we acknowledge that extending the methodology to other countries and explicitly linking the resulting archetypes to risk metrics or impact data (e.g., disaster losses, preparedness levels, or recovery times) would be necessary to fully move from a descriptive typology to one with predictive and policy-relevant implications. The authors have clarified this point in the revised manuscript (see response to comment 2a and 2c), and we frame our current analysis as a foundational step that enables such future cross-country, risk-informed comparisons.

    d.  L737-738: The sentence "This study demonstrates the potential of urban archetypes in developing risk storylines, enhancing risk communication, and supporting stakeholder engagement" is simply not true. Where exactly in the manuscript are these three points demonstrated?

**Response:** Section 6.1 has been added to underline the usefulness of this study in decision-making process. This section presents a simplified impact susceptibility index assigned to each archetype, based on their demographic, socio-economic and geographic characteristics, that reflect their vulnerability and exposure level. See also response to comment 2a. To underly this point, the paragraph dedicated to application validity dimensions in the Discussion section has been modified as follows:

*"Application Validity evaluates the practical usefulness of the archetypes. This dimension can be addressed emphasizing practical applications of archetypes in policy, planning, and governance, for instance, by presenting results to government officials and researchers, guiding inform local policy discussions, with archetypes guiding differentiated policy interventions (Nagel et al., 2024). Section 6.1 illustrates the potential of urban archetypes to enhance risk communication through the assignment of a simplified impact susceptibility index to each identified archetype. Additionally, the exploratory case study presented in Marciano et al. (2024) highlights how these archetypes can support stakeholder engagement by informing the development of multi-risk storylines. By categorizing urban areas into distinct archetypes, it becomes possible to assess how different hazard scenarios may unfold in each context, considering their specific vulnerabilities, exposure levels, and adaptive capacities. This structured approach enables policymakers to design tailored interventions and resilience strategies based on specific vulnerability profiles. However, to further strengthen urban resilience planning and develop targeted mitigation measures, it is crucial to consider not only exposure and vulnerability but also hazard data for each archetype—particularly the level of exposure of urban settlements to various natural hazards. Although in this study we did not yet integrate hazard information, there is a clear need for future research to incorporate this aspect and conduct GIS-based analyses for a more comprehensive assessment of urban risk (e.g., Tocchi et al., 2024)."*

7.  In the last paragraph of the conclusions (L759-761): "its findings offer a valuable tool for policy makers to design targeted interventions based on specific vulnerability profiles": The paper does not demonstrate that the archetypes are indeed associated with specific vulnerability profiles, it just assumes they are. Again, the paper needs to somehow show that the archetypes are indeed associated with vulnerability profiles that make sense for policy making.

**Response:** The authors fully acknowledge the need to demonstrate, rather than assume, that the identified archetypes correspond to specific and meaningful vulnerability profiles. The carefully selection of indicators based on both theoretical relevance and empirical evidence from the literature linking them to disaster risk and vulnerability (e.g., demographic pressure, social fragility, accessibility to services, and urban density) used for clustering ensure that each archetype is indeed associated with specific and different vulnerability profiles. The carefully selection of indicators also fully satisfy the construction validity dimension of archetype analysis (e.g.,Diogo et al., 2023; Nagel et al., 2024). To further demonstrate that each archetype is associated with a distinct vulnerability profile, we developed a simplified impact susceptibility index (see responses

to Comments 2a and 6d). This index aggregates the key vulnerability-related indicators used in the clustering process, allowing us to rank archetypes according to their average vulnerability and exposure levels. The results show that archetypes differ significantly in their composite index scores, supporting the interpretation that they represent distinct profiles of susceptibility to hazard impacts. Still, the limits of this study in fully satisfy empirical evidence of archetypes has also been addressed (see response to comment 2a).

Minor Comments/Edits

1. L13: "complexity in an otherwise too broad problem".

   **Response:** The text has been modified from "helping to segmentate the complexity in otherwise too broad problem" to "helping to segmentate the complexity in an otherwise too broad problem".

2. L24: "Over the last few decades".

   **Response:** The text has been modified from "Over last few decades" to "Over the last few decades".

3. L43: Exposure is not "intended". Do you mean understood? Defined?

   **Response:** The text has been modified as follows: "*Exposure refers to the presence of people, livelihoods, species or ecosystems, environmental functions, services, and resources, infrastructure, or economic, social, or cultural assets in places and settings that could be adversely affected, while vulnerability refers to the propensity or predisposition to be adversely affected*"

4. L43-45: Please replace the semicolons ( ; ) by commas.

   **Response:** The text has been modified. Please, see the response to the previous comment.

5. L58: It says "we assume", but perhaps you mean "we hypothesize"?

   **Response:** In this case, "we assume" was deliberately chosen, as it reflects the starting premise or working assumption of the study rather than a formal, testable hypothesis. The use of "assume" emphasizes that this is an underlying premise guiding the methodological approach — that a limited number of archetypes can sufficiently represent the diversity of urban vulnerability and exposure profiles at regional and national scales. Therefore, we prefer to maintain the use of "we assume" in this context.

6. L61: Erase "We note that". It is not necessary.

   **Response:** The second part of the introduction section has been totally revised as suggested by reviewers, in order to: (i) clearly stating the research question early on; (ii) elaborating on how archetypes enhance the understanding of exposure and vulnerability; (iii) clarifying different interpretations of archetypes and explicitly stating the one used in this study; (iv) briefly explaining how the concept is operationalized in this work. Specifically, lines from 58 to 68 have been modified as follows:

   *"This study addresses the following research question: can urban settlements be clustered into meaningful archetypes based on shared characteristics of vulnerability and exposure, to improve multi-risk assessment and support more targeted resilience planning at regional and national scales? Indeed, despite the high specificity of exposure and vulnerability of each urban and rural environment, we assume that a relatively low number of representative archetypes could be found to decrease the level of complexity at regional and national scale, ensure a baseline for comparison and highlight potential hotspots in multi-hazard and multi-risk assessment frameworks.*

   *The term "archetype" can be interpreted in different ways. In statistics, archetypes refer to extremal profiles used to describe all data points as convex combinations of a few "pure" types* (Cutler & Breiman, 1994) *. In contrast, in sustainability science and climate risk research, archetypes are understood as representative specimens or clusters of similar entities that are "crucial for describing the system dynamics or causal effect of interest" and that exhibit recurring patterns of risk-relevant characteristics " (Oberlack et al., 2019). We*

*adopt this latter interpretation. In our work, urban and rural settlement archetypes are defined as representative instances (real or ideal) of a group of municipalities sharing similar vulnerability and exposure characteristics."*

7. L83-85: Are capitals really needed for "design", "analysis" and "application"?

   **Response:** As suggested, capital letters have been removed.

8. L95: No comma after "defined".

   **Response:** As suggested, comma has been removed.

9. Table 1: It says "GPD per capita" instead of "GDP per capita".

   **Response:** In Table 1 the text "*GPD per capita*" has been corrected to "*GDP per capita*".

10. L161: "complicates evacuation efforts and strains emergency response resources".

    **Response:** Text has been modified from "*complicates evacuation efforts and strain emergency response resources*" to "*complicates evacuation efforts and strains emergency response resources*".

11. L167: Consider breaking the paragraph here and starting a new one to talk about the socio-economic factors.

    **Response:** As suggested, a break has been introduced between the paragraph describing the demographic parameters and the one describing the socio-economic parameters.

12. L169: "highlight that the elderly".

    **Response:** The text has been modified from "*highlight that elderly*" to "*highlight that the elderly*".

13. L192: "proposed a grid-based approach".

    **Response:** The text has been modified from "*proposed grid-based approach*" to "*proposed a grid-based approach*".

14. L201: "adopting to the abovementioned" (remove "to").

    **Response:** The text has been modified from: "*classification of municipalities adopting to the above-mentioned Eurostat procedure is provided by ISTAT*" to "*classification of municipalities adopting the above-mentioned Eurostat procedure is provided by ISTAT*".

15. Fig. 1: Please consider changing the colour of peri-urban areas. The map on the right shows six labels, but only three classes of urban centeredness degree are considered in the analysis. "Hub" and "municipal hub" can be seen as different shades of red of the final class "urban hubs", while the intermediate, peripheral and ultra-peripheral inland areas are indifferent shades of blue, so that is clear. However, peri-urban areas are in orange, which makes them very similar to the "urban hubs" class. A different colour for peri-urban areas would make it easier to visualise the final three classes used. Parentheses in the text (lines 243-245) could clarify this. E.g. "namely urban hubs (represented…, shades of red in Fig. 1), peri-urban areas (new colour) and inland areas (shades of blue)".

    **Response:** Thanks for this comment. As suggested, Figure 1 has been modified changing the colour of peri-urban areas (from light orange to green). Moreover, a clarification about colour used in the map has been added in the text: "*namely urban hubs (represented by both hubs and intermunicipal hubs, **shades of red in Figure 1**), peri-urban areas (**green in Figure 1**) and inland areas (that includes intermediate, peripherical and ultra-peripherical areas, **shades of blue in Figure 1**)*".

    The new map is reported below:

[Figure]

Legend:
- Hub
- Intermunicipal hub
- Peri-urban area
- Intermediate area
- Peripheral area
- Ultra-peripheral area

16. L258: "lead to the model becoming".

    **Response:** The text has been modified from "*lead the model becoming*" to "*lead to the model becoming*"

17. L259: "correlation that may exist".

    **Response:** In the sentence "*Furthermore, despite some correlation may exist between urban vulnerability and coastal/inland areas*..." the term "*despite*" should be directly followed by a noun, gerund (-ing form), or pronoun, not a full clause with "*that may exist*". Therefore, the mentioned sentence has been modified as: "*Furthermore, despite some correlation existing between urban vulnerability and coastal/inland areas*...".

18. Table 2 vs line 377 and Fig. 3: Is it population aged under 15 or 14 (different numbers in different places).

    **Response:** The correct variable is the population under 15 (not under 14). The text in line 377 and Figure 3 has been modified accordingly.

19. Table 2 vs line 311: Should it be "total population aged over 30" under "high educational index"?

    **Response:** The high educational index is calculated as people with at least a university degree compared to the total population aged over 30, as mentioned in the text (line 311). There was a type in Table 2 (table that has been removed after the revision process; see response to comment 3).

20. L344: Clustering methods per se do not "ensure" that observations within the same group are highly similar. It is their goal, but it is not guaranteed. They are just numerical methods. They will return some sort of clustering no matter what you provide as input. Please re-phrase.

    **Response:** To correctly reflect that clustering methods seek to achieve internal similarity and external separation, but do not guarantee it, the sentence: "*This method ensures that observations within the same group are highly similar, while those in different groups are distinctly different*" has

been changed to "*This method is designed to group together observations that are highly similar, while separating those that differ into distinct clusters*"

21. L354-355: "More detailed information".

    **Response:** The text "*Most detailed information*" has been replaced by "*More detailed information*".

22. L359: "is crucial for enhancing the quality of the clustering".

    **Response:** The text "*Data preprocessing is crucial for enhance quality of clustering*" has been replaced by "*Data preprocessing is crucial for enhancing the quality of clustering*".

23. L376-383: The strong negative correlation between the proportion of employed and un

    **Response:** In the previous lines of the same paragraph (lines 374-375) it is already specified that "*The value of r ranges between -1 and 1, with values higher than 0 that indicate a positive correlation and values lower than 0 a negative correlation*." Thus, there is no need to further specify whether the correlation is negative or positive in lines 376-383, the sign of *r* is clearly provided for each pair of variables.

24. Fig. 3, labels (list 1-14):
    a. "Proportion of population under 14" (or another alternative, the current grammar is not right).
    b. "Proportion of population over 65" (or another alternative, the current grammar is not right).
    c. Replace "components" with "members".
    d. "Residents" (plural), not "resident" (singular).

    In general, please aim for consistency between the labels used in this figure and in table 3.

    **Response:** As suggested, the labels have been modified. Please, see the response to the following comment which provides the new version of figure 3 with corrected labels.

25. Fig. 3: It would be easier to interpret the matrix if the colour scale was based on three colours instead of two (right now, from yellow to blue, passing through green). This would make values around zero more visible.

    **Response:** Colors of the figure have been modified, in order make it easier to interpret also for readers with color blindness. Specifically, a blue-white-red color scheme has been adopted to improve accessibility. Please, find below the figure with new color scheme.

[Figure]

1. Proportion of population under 14
2. Proportion of population over 65
3. Aging index
4. Dependency ratio
5. High educational index
6. Low educational index
7. Proportion of employed
8. Proportion of unemployed
9. Proportion of commuters
10. Proportion of female employed
11. Proportion of buildings in poor conditions
12. Proportion of families with 5 or more members
13. Crowding index
14. Proportion of foreign residents

26. Fig. 3: Is it correct that the correlation between features 7 and 8 is so low, numerically? These features are the proportion of employed and unemployed among the working-age population. Are they not one minus the other (1 – employed = unemployed) and, if not (because of some other category), are they not at least highly negatively correlated?

**Response:** In Italy, according to ISTAT definitions, the number of employed refers to people aged over 15 who are actively working. However, the category of "working-age population" (people aged over 15) also includes individuals who are not in the labor force, such as students, retirees, housewares and others not seeking employment. Therefore, 1 – employed ≠ unemployed. For this reason, the correlation between the proportion of employed and unemployed among the working-age population is not necessarily strongly negative.

In the revised version of the paper, when presenting social vulnerability variables (section 3.5), it has been specified that working-age populations adopted for the calculation of the employment/unemployment rate is the population aged over 15.

27. L406: Should the subscript of the x variables be "l" instead of "k", to be consistent with Eq. (1) and the text?

**Response:** Yes, it is correct. The subscripts of the x variables have been corrected using the right letter ("l" instead of "k").

28. L515: "The attributes' importance for clustering is evaluated adopting the simplified procedure".

**Response:** The text "*Attributes importance for clustering is evaluated adopting simplified procedure*" has been replaced by "The attributes' importance for clustering is evaluated adopting the simplified procedure".

29. L516: "to assess their contribution".

**Response:** The text "to assess its contribution" has been replaced by "to assess their contribution".

30. L519: "WCD as the performance metric".

**Response:** The text "In this study we consider WCD as performance metric" has been replaced by "In this study we consider WCD as the performance metric".

31. L526: Do you mean cluster 2, instead of 6?

**Response:** Yes, it was a typo. In the text, "Cluster 6" has been replaced by "Cluster 2".

32. L564: "The lower is the WCDl" (erase "is").

**Response:** The text "The lower is the WCDl" has been replaced by "The lower the WCDl".

33. Table 4 is colour-coding the SoVI. Please mention the colours in the text (lines 614-616).

**Response:** The description of color code used to rank SoVI value has been added in the text. Specifically, lines 613-617 have been modified as follows (added text is reported in bold):

*"The criteria used to identify the different socio-economic condition categories is based on SoVI values and specifically: a value lower than 1 corresponds to low social vulnerability **(dark green in Table 4),** value between 1 and 1.20 to moderate social vulnerability **(light green in Table 4),** values between 1.20 and 1.40 to intermediate social vulnerability **(yellow in Table 4),** values between 1.40 and 1.60 to high social vulnerability **(light red in Table 4),** values higher than 1.60 to very high social vulnerability **(dark orange in Table 4)**. The average values of individual variables for each sub-cluster are provided in Table 4."*

34. Table 4:
    a. "Building poor state"? You used "poor" before. Please be consistent.
    b. "Foreigners" or "Foreign residents".
    c. Clusters 2a and 4b: "Settlements with high social vulnerability".

    **Response:** The suggested corrections have been applied. Specifically:

    a. "*Buildings bad state*" has been replaced by "*Buildings poor state*"
    b. "*Foreign*" has been replaced by "*Foreigns*"
    c. "*High socially vulnerability settlements*" has been replaced by " *Settlements with high social vulnerability*" for Clusters 2a, 3c, 4b and 10a. Accordingly, Cluster 8a has been renamed as "*Settlements with very high social vulnerability*" and Cluster 9a as "*Settlements with low social vulnerability*".

35. Table 4: As they are phrased, sub-clusters 2a (settlements with high social vulnerability) and 2b (settlements with aged population with high social vulnerability) overlap, with 2b seemingly a sub-category of 2a. If 2a explicitly excludes aged populations, then the label should reflect this

    **Response:** As mentioned in the text, sub-cluster names may also incorporate the specific social vulnerability factors that contribute most significantly to the SoVI value. Therefore, since sub-cluster 2a shows the highest average crowding index, it has been renamed as "*High household density settlements with high social vulnerability*".

36. L696-697: "in the southern regions, with almost all cases located".

    **Response:** The text "*in the southern regions, which almost all cases located in Campania*" has been replaced by "*in the southern regions, with almost all cases located*".

*References*

[revised manuscript text omitted]

Zuzak, C., Mowrer, M., Goodenough, E., Burns, J., Ranalli, N., & Rozelle, J. (2022). The national risk index: establishing a nationwide baseline for natural hazard risk in the US. *Natural Hazards*, *114*(2), 2331–2355. https://doi.org/10.1007/s11069-022-05474-w

---

## Referee Report (RR1)

**Review of Manuscript egusphere-2025-908 R1**

I thank the authors for their detailed replies to my previous comments and for having taken them into account in the revised version of the manuscript. While the lack of confrontation of the proposed archetypes against risk data or models that demonstrate the relevance of the classification is still a shortcoming, the more in-depth discussion of this limitation as well as the new section on the vulnerability profiles have contributed to improve the quality of the paper. I recommend this work be published after addressing the following minor comments and edits.

**Minor Comments**

1. L190-192, L205-206, L413-415, Tables 1 & 2, Fig. 3, etc.: While the authors have improved the presentation of the variables they considered with respect to the original manuscript, it is still not straightforward to navigate this matter throughout the manuscript. Section 2 should be more explicit regarding the fact that a larger number of indicators were considered for inclusion but only a sub-set was used, and enumerating the reasons for doing so (e.g., correlation analysis, availability). For example:

   a) "Dependency ratio" is mentioned in L190, Table 1, Fig. 3 and the lines before it, but then it is dropped. Only in the analysis around Fig. 3 it becomes clear that the variable was considered but then discarded due to it being strongly correlated to other variables that were selected in the end. But when reading L190 it gives the impression of being used.

   b) The indicator names in L190-192 is different from that in Table 2, and line 192 directly refers to this table. This makes it very confusing for the reader, who might also wonder why the dependency ratio is not in the table (if they go and see the table when directed to do so in L192, before reading the correlation analysis).

   c) L205 mentions 19 attributes, Table 2 includes 12 variables, Fig. 3 includes 14. This is confusing for the reader if it is not accompanied by an explanation early on in the text (i.e., Section 2).

   Perhaps it would help to add text around certain lines to clarify early on to the reader that not all initially-considered indicators were actually used for the clustering analysis. For example, L190-192 could read: "The complete list of socio-economic indicators considered includes age, dependency ratio, … […] … and percentage of women in the workforce, though some indicators were not used for the final clustering process due to their strong correlation with other selected variables (Table 2)", or similar. I am not saying that the details of Section 3.1 should be stated earlier, but the overall strategy (considered a large number of indicators initially, made a selection of a sub-set to be used for the clustering process) should.

2. L744-747: If I am understanding correctly, the composite index of Sibilia et al. (2024) is not used by the authors. If so, I recommend removing the lines "In Sibilia et al… […] … and environmental", as they include too many details about an index that is irrelevant to what follows.

3. L767, Fig. 10: Several points here:

   a) The figure and discussion focus on the average ISI but no dispersion is reported. The dispersion is relevant to understand how different the results for each archetype are from one another, especially because so many of them have very similar mean ISI values. Please consider including some sort of measure of dispersion (whichever is appropriate to represent the numerical results) and include it in the discussion.

   b) It is interesting that the three sub-clusters of cluster 3 have very similar ISI values. Their SoVI values are similar as well (Table 3). Are the ISI dispersion values similar as well? Are there any interesting insights to the method or the nature of these archetypes that are worth discussing in the text?

   c) Following up from the previous point, looking at certain archetypes one gets the impression that the mean ISI values are highly correlated with the mean SoVI values (Table 3). For example, the three subclusters if cluster 3, as mentioned above, the fact that cluster 1 has the lowest ISI and one of the lowest SoVI values (albeit not the lowest), the fact that subclusters 2a and 2b have very similar ISI values, larger than subcluster 2c, and present the same pattern in their SoVI values, the fact that 8a has both the highest ISI and SoVI

values, etc. However, other archetypes suggest the opposite. For example, subclusters 9a and 9b have opposite trends of ISI vs SoVI, cluster 6 has one of the lowest SoVI values but not so much regarding ISI, etc. It would be interesting and insightful to explore these and similar observations and comment about their potential meaning and significance, if any.

4. L880: In lines 824-825 you discuss as well the limitations of risk modelling for the purpose of validating the archetypes. I suggest adding this to the sentence "due to the lack of fully integrated social and institutional vulnerability data, as well as limitations of risk modelling".

**Edits**

1. L104: Add a comma after "first" (i.e., "first, broad urban and rural…".

2. L167: Erase "acknowledging that".

3. L168: Add "the" between "factors of" and "built environment".

4. L175: "Greater population" instead of "Higher population".

5. L177: Consider starting a new paragraph at "The degree of urbanisation…".

6. L242: Should it say three categories, instead of four?

7. L262-264: "minimize", "prevent" and "enhance". These three items follow the sentence "in order to (infinitive verb)".

8. L404-411: The text says "proportion of under 15 aged" while Fig. 3 says "proportion of population under 14". Is it 14 or 15? Moreover, please change "proportion of under X aged" (this is grammatically incorrect) into "proportion of population under X" or "proportion of age X and under".

9. L418, Table 2: I infer the variable "Population class" refers to section 2.4. If so, consider rephrasing as "Residential population class" or similar, to increase consistency with the main text.

10. All throughout, "urban degree" does not read well. Replace with "degree of urbanisation".

11. All throughout, "urban centredness degree" does not read well. Replace with "degree of urban centredness".

12. L667, Fig. 9: Introducing the map with the Italian regions is a useful addition. However, the resolution of the image is quite poor.

13. L877: "empirical and external validity were only partially addressed" (not "was").

14. L877: Broken link…? "Fare clic…"?

---

## Author Response (AR2)

**Review of Manuscript egusphere-2025-908R1**

I thank the authors for their detailed replies to my previous comments and for having taken them into account in the revised version of the manuscript. While the lack of confrontation of the proposed archetypes against risk data or models that demonstrate the relevance of the classification is still a shortcoming, the more in-depth discussion of this limitation as well as the new section on the vulnerability profiles have contributed to improve the quality of the paper. I recommend this work be published after addressing the following minor comments and edits.

**Response**: The authors thank the reviewer for their additional comments and suggestions. Please find below the responses to each comment, along with the corresponding edits made to the main text.

**Minor Comments**

1. L190-192, L205-206, L413-415, Tables 1 & 2, Fig. 3, etc.: While the authors have improved the presentation of the variables they considered with respect to the original manuscript, it is still not straightforward to navigate this matter throughout the manuscript. Section 2 should be more explicit regarding the fact that a larger number of indicators were considered for inclusion but only a sub-set was used, and enumerating the reasons for doing so (e.g., correlation analysis, availability). For example:

   a) "Dependency ratio" is mentioned in L190, Table 1, Fig. 3 and the lines before it, but then it is dropped. Only in the analysis around Fig. 3 it becomes clear that the variable was considered but then discarded due to it being strongly correlated to other variables that were selected in the end. But when reading L190 it gives the impression of being used.

   **Response**: To clarify which indicators were adopted in the study, the original sentence: 'The complete list of socio-economic indicators considered includes age, dependency ratio, level of education, family structures, commuting rate, quality of buildings, race/ethnicity, employment rate, percentage of women in the workforce (Table 2)' (lines 190–192), has been replaced with: "*A comprehensive list of socio-economic indicators considered is presented in section 2.5, though some indicators were not used for the final clustering process due to their strong correlation with other selected variables (Table 2).*"

   b) The indicator names in L190-192 is different from that in Table 2, and line 192 directly refers to this table. This makes it very confusing for the reader, who might also wonder why the dependency ratio is not in the table (if they go and see the table when directed to do so in L192, before reading the correlation analysis).

   **Response**: Lines 190–192 have been revised, as indicated in the response to the previous comment.

   c) L205 mentions 19 attributes, Table 2 includes 12 variables, Fig. 3 includes 14. This is confusing for the reader if it is not accompanied by an explanation early on in the text (i.e., Section 2).

   **Response**: The sentence in line 205 ("*The dataset includes 7960 objects, representing the 7960 Italian municipalities, and 19 attributes (both numerical and categorical) related to the vulnerability factors outlined in sections 3.1 through 3.5*") has been revised to "*All data are collected at the municipal level, aligning with the administrative boundaries adopted for the*

*analysis. The dataset includes 7960 objects, representing the 7960 Italian municipalities, and the numerical and categorical attributes related to the vulnerability factors outlined in sections 3.1 through 3.5.*". Additionally, to improve clarity regarding the indicators ultimately used for clustering, the following sentence has been added at line 333: "*It is important to note that only 12 of the 14 previously presented social vulnerability indicators are used in this study (Table 2), as a correlation analysis - described in Section 3.1 - was conducted*". Finally, the sentence in line 405 has been modified to (edits reported in orange): "Figure 3 shows the correlation matrix obtained for the 14 numerical variables presented in section 2.5."

Perhaps it would help to add text around certain lines to clarify early on to the reader that not all initially-considered indicators were actually used for the clustering analysis. For example, L190-192 could read: "The complete list of socio-economic indicators considered includes age, dependency ratio, ... [...] ... and percentage of women in the workforce, though some indicators were not used for the final clustering process due to their strong correlation with other selected variables (Table 2)", or similar. I am not saying that the details of Section 3.1 should be stated earlier, but the overall strategy (considered a large number of indicators initially, made a selection of a sub-set to be used for the clustering process) should.

**Response**: Lines 190–192 have been revised in response to the reviewer's comment, as outlined in the response to Comment 1a

2. L744-747: If I am understanding correctly, the composite index of Sibilia et al. (2024) is not used by the authors. If so, I recommend removing the lines "In Sibilia et al... [...] ... and environmental", as they include too many details about an index that is irrelevant to what follows.

    **Response**: The lines citing the composite index from Sibilia et al. have been removed.

3. L767, Fig. 10: Several points here:

    a) The figure and discussion focus on the average ISI but no dispersion is reported. The dispersion is relevant to understand how different the results for each archetype are from one another, especially because so many of them have very similar mean ISI values. Please consider including some sort of measure of dispersion (whichever is appropriate to represent the numerical results) and include it in the discussion.

    **Response**: The following figure (10b) showing a boxplot of ISI values has been added:

[Figure]

In section 6.1 the following discussion, related to this figure, has been added:

*"The box plot in Figure 10b illustrates the distribution of ISI values across the identified urban archetypes, providing insights into both central tendency and internal variability. Archetype 1 exhibits the lowest median ISI and minimal variability, suggesting a consistently low level of exposure and vulnerability across its municipalities. In contrast, Archetype 8a displays the highest median ISI with a very narrow spread, indicating strong internal homogeneity and high susceptibility to impacts. The few outliers with an ISI value of 9 highlight minor deviations but do not significantly affect the overall pattern. Archetype 9b shows the greatest dispersion, reflecting a high degree of internal heterogeneity. This wide variability suggests the presence of municipalities with both relatively low and high ISI values within the same archetype, potentially complicating uniform policy interventions. Archetypes 3a, 3b, and 3c present identical median, mean, and interquartile ranges, which aligns with their shared geographic and demographic features, as well as similar SoVI scores (see Table 3). However, their comparable SoVI outcomes result from distinct socio-economic compositions, as discussed in Section 5, underscoring the multidimensional nature of social vulnerability."*

b)  It is interesting that the three sub-clusters of cluster 3 have very similar ISI values. Their SoVI values are similar as well (Table 3). Are the ISI dispersion values similar as well? Are there any interesting insights to the method or the nature of these archetypes that are worth discussing in the text?

**Response:** Archetypes 3a, 3b, and 3c exhibit very similar ISI values due to their shared demographic and geographic characteristics, as well as comparable SoVI scores. However, these similar SoVI values result from different underlying socio-economic features. For example, Archetype 3a has a higher mean aging index, while Archetype 3b is characterized by a higher crowding index. A note on this has been added to the manuscript (see response to the previous comment). Additionally, the following sentence has been included in Section 5 to clarify the value of defining individual archetypes that may reflect specific social vulnerability factors: *"Understanding the influence of individual socio-economic indicators within each archetype can support the prioritization and tailoring of risk mitigation strategies and resilience policies"*.

c)  Following up from the previous point, looking at certain archetypes one gets the impression that the mean ISI values are highly correlated with the mean SoVI values (Table 3). For example, the

three subclusters if cluster 3, as mentioned above, the fact that cluster 1 has the lowest ISI and one of the lowest SoVI values (albeit not the lowest), the fact that subclusters 2a and 2b have very similar ISI values, larger than subcluster 2c, and present the same pattern in their SoVI values, the fact that 8a has both the highest ISI and SoVI values, etc. However, other archetypes suggest the opposite. For example, subclusters 9a and 9b have opposite trends of ISI vs SoVI, cluster 6 has one of the lowest SoVI values but not so much regarding ISI, etc. It would be interesting and insightful to explore these and similar observations and comment about their potential meaning and significance, if any.

**Response:** The following text has been added to the discussion to account for the influence of social vulnerability:

*"In several cases, social vulnerability is the primary driver of high ISI values, as observed in Archetypes 8a, 10a, 7 and Archetypes 2a, 2b and 2c. The latter (i.e., Archetypes 2a, 2b, and 2c) share the same geographic and demographic profiles yet differ in ISI values - with 2a and 2b showing higher ISI than 2c - due solely to differences in their SoVI scores. In this context, SoVI emerges as the only influencing factor driving ISI variation among these archetypes. Conversely, for other archetypes, demographic and geographic characteristics play a more significant role in shaping ISI outcomes. For instance, Archetype 9b presents the lowest SoVI but a relatively high ISI, which can be attributed to its high population density. In contrast, Archetype 3b shows one of the highest mean SoVI scores, second only to Archetype 9b, but results in a relatively low ISI, primarily due to its low population density and geographic remoteness."*

4. L880: In lines 824-825 you discuss as well the limitations of risk modelling for the purpose of validating the archetypes. I suggest adding this to the sentence "due to the lack of fully integrated social and institutional vulnerability data, as well as limitations of risk modelling".

**Response:** The sentence in line 880 has been revised as suggested.

**Edits**

1. L104: Add a comma after "first" (i.e., "first, broad urban and rural…".

2. L167: Erase "acknowledging that".

3. L168: Add "the" between "factors of" and "built environment".

4. L175: "Greater population" instead of "Higher population".

5. L177: Consider starting a new paragraph at "The degree of urbanisation…".

6. L242: Should it say three categories, instead of four?

7. L262-264: "minimize", "prevent" and "enhance". These three items follow the sentence "in order to (infinitive verb)".

8. L404-411: The text says "proportion of under 15 aged" while Fig. 3 says "proportion of population under 14". Is it 14 or 15? Moreover, please change "proportion of under X aged" (this is grammatically incorrect) into "proportion of population under X" or "proportion of age X and under".

9. L418, Table 2: I infer the variable "Population class" refers to section 2.4. If so, consider rephrasing as "Residential population class" or similar, to increase consistency with the main text.

10. All throughout, "urban degree" does not read well. Replace with "degree of urbanisation".

11. All throughout, "urban centredness degree" does not read well. Replace with "degree of urban centredness".

12. L667, Fig. 9: Introducing the map with the Italian regions is a useful addition. However, the resolution of the image is quite poor.

13. L877: "empirical and external validity were only partially addressed" (not "was").

14. L877: Broken link…? "Fare clic…"?

**Response:** All suggested edits have been implemented.